# Improving Local Effectiveness for Global robust training

## Abstract

Despite its popularity, deep neural networks are easily fooled. To alleviate this deficiency, researchers are actively developing new training strategies, which encourage models that are robust to small input perturbations. Several successful robust training methods have been proposed. However, many of them rely on strong adversaries, which can be prohibitively expensive to generate when the input dimension is high and the model structure is complicated. We adopt a new perspective on robustness and propose a novel training algorithm that allows a more effective use of adversaries. Our method improves the model robustness at each local ball centered around an adversary and then, by combining these local balls through a global term, achieves overall robustness. We demonstrate that, by maximizing the use of adversaries via focusing on local balls, we achieve high robust accuracy with weak adversaries. Specifically, our method reaches a similar robust accuracy level to the state of the art approaches trained on strong adversaries on MNIST, CIFAR-10 and CIFAR-100. As a result, the overall training time is reduced. Furthermore, when trained with strong adversaries, our method matches with the current state of the art on MNIST and outperforms them on CIFAR-10 and CIFAR-100.

## 1 Introduction

With the proliferation of deep neural networks (DNN) in areas including computer vision, natural language processing and speech recognition, there has been a growing concern over their safety. For example, Szegedy et al. (2013) demonstrated that naturally trained DNNs are in fact fragile. By adding to each data a perturbation that is carefully designed but imperceptible to humans, DNNs previously reaching almost $100\%$ accuracy performance could hardly make a correct prediction any more. This could cause serious issues in areas such as autonomous navigation or personalised medicine, where an incorrect decision can endanger life. To tackle these issues, training DNNs that are robust to small perturbations has become an active area of research in machine learning.

Various algorithms have been proposed (Papernot et al., 2016; Kannan et al., 2018; Zhang et al., 2019b; Qin et al., 2019; Moosavi-Dezfooli et al., 2020; Madry et al., 2018; Ding et al., 2020). Among them, adversarial training (ADV) (Madry et al., 2018) and TRADES (Zhang et al., 2019b) are two of the most frequently used training methods so far. Although developed upon different ideas, both methods require using strong adversarial attacks, generally computed through several steps of projected gradient descent. Such attacks can quickly become prohibitive when model complexity and input dimensions increase, thereby limiting their applicability. Since the cost of finding strong adversaries is mainly due to the high number of gradient steps performed, one potential approach to alleviate the problem is to use cheap but weak adversaries. Weak adversaries are obtained using fewer gradient steps, and in the extreme case with a single gradient step. Based on this idea, Wong et al. (2020) argue that by using random initialization and a larger step-size, adversarial training with weak adversaries found via one gradient step is sufficient to achieve a satisfactory level of robustness. We term this method as one-step ADV from now on. While one-step ADV does indeed exhibit robustness, there is still a noticeable gap when compared with its multi-step counterpart.

In this paper, we further bridge the gap by proposing a new robust training algorithm: Adversarial Training via LocAl Stability (ATLAS). Local stability, in our context, implies stability of prediction and is the same as local robustness. Specifically, we make the following contributions:

- We adopt a new perspective on robust accuracy and introduce a framework for constructing robust training losses that allow more effective use of adversaries. The framework consists of a local component and a global component. The local component maximizes the effectiveness of an given adversary by improving the network's robustness on both the adversary and points around it. In other words, the local component attempts to increase the radius of a ball centered at the adversary on which the network is being robust. The global component combines all local balls in a regularized way to achieve the desired overall robust performance.

- Based on the framework and guided by the need of fast robust training, we propose our novel robust training algorithm ATLAS.

- We show that ATLAS makes a more effective use of weak adversaries by favourably comparing it against one-step ADV on three datasets: MNIST, CIFAR-10 and CIFAR-100.

- Although one-step ATLAS is more expensive than its other one-step counterparts, ATLAS still allows efficient robust training. We show that, with a one-step weak adversary, ATLAS manages to achieve comparable levels of robust accuracy to multi-step state of the art methods on all datasets.

- Finally, we show that when strong adversaries are used, ATLAS matches with the current state of the art on MNIST and outperforms them on CIFAR-10 and CIFAR-100.

## 2  RELATED WORKS

Robust training aims to learn a network such that it is able to give the same correct output even when the input is slightly perturbed. Existing robust training algorithms can be divided into two categories: natural image based methods and adversaries based methods. Within the first category, the common form of loss is a natural loss term plus a regularizer computed at natural images. We briefly mention some of these methods. Moosavi-Dezfooli et al. (2020) observed empirically that reducing the curvature of the loss function and the decision boundary could lead to robust models. The authors thus propose a regularizer based on the Hessian of the loss function. Closely related is the regularizer, introduced in (Jakubovitz & Giryes, 2018; Hoffman et al., 2019; Ross & Doshi-Velez, 2018), that penalizes the Frobenius norm of the Jacobian of the loss function. Jacobian regularizer can also be seen as a way of reducing the curvature of the decision boundary (Jakubovitz & Giryes, 2018). Although calculating the norm is computationally expensive, a fast estimator with empirically high accuracy has been developed in Hoffman et al. (2019).

We focus on adversary based robust training, as they generally perform better in terms of robust accuracy. Under this category, an early fundamental work is the Fast Gradient Sign Method (FGSM) by Goodfellow et al. (2015). Adversarial Training (ADV) (Madry et al., 2018) is a multi-step variant of FGSM. Rather than using one step FGSM, ADV employs multi-step projected gradient descent (PGD) (Kurakin et al., 2016) with smaller step-sizes to generate perturbed inputs. These modifications have allowed ADV to become one of the most effective robust training methods so far (Athalye et al., 2018). Another frequently used robust training method is TRADES (Zhang et al., 2019b). Motivated by its theoretical analyses of the trade-off between natural accuracy and robust accuracy in a binary classification example, TRADES encourages model robustness by adding to the natural loss a regularizer involving adversaries to push away the decision boundary.

Recently, Qin et al. (2019) suggest that a robust model can be learned through promoting linearity in the vicinity of the training examples. They designed a local linearity regularizer (LLR), in which adversaries are used to maximize the penalty for non-linearity. Applying LLR also allows efficient robust training. We note that the underlying idea of LLR is complementary to ATLAS. In addition, several works have suggested to adopt input dependent treatments. These works include incorporating the fact of whether the given input is correctly classified (Wang et al., 2020) and using adaptive perturbation for different inputs (Ding et al., 2019).

One major drawback of adversary based methods (Madry et al., 2018; Zhang et al., 2019b; Qin et al., 2019; Wang et al., 2020; Ding et al., 2019) is that most of them rely on strong adversaries, computed via expensive PGD. When the input dimension is high and the model structure is complicated, finding adversaries can be too expensive for these methods to work effectively. Several works have researched possible ways to speed up the process. Algorithmically, Zhang et al. (2019a) cut down the total

number of full forward and backward passes for generating multi-step PGD adversaries while Zhang et al. (2020) introduce a parameter to allow early stop PGD. Shafahi et al. (2019) reduce adversary overhead by combining weight update and input perturbation update within a single back propagation and use a single step FGSM adversary. Wong et al. (2020) argue that the main reason one-step FGSM, generally regarded as not robust, is effective in (Shafahi et al., 2019) is due to the non-zero initialization used. As a result, Wong et al. (2020) proposed that with random initialization and a larger step size, weak adversaries generated by FGSM could lead to models with a high level of robust accuracy. In this study, we adopt a similar viewpoint of accelerating robust training through the use of weak adversaries.

## 3 PRELIMINARIES AND NOTATIONS

We consider classification tasks. Let $\boldsymbol{x}$ be an image from the data distribution $\mathcal{X} \subset \mathbb{R}^N$ and $y_{\boldsymbol{x}}$ be its correct label from classes $\mathcal{C} = \{1, \ldots, C\}$. We denote a neural network, parameterized by $\boldsymbol{\theta}$, as $f(\boldsymbol{x}; \boldsymbol{\theta}) : \mathcal{X} \to \mathbb{R}^C$. The function $f$ outputs logits and the predicted class label is chosen to be the element with the largest scalar value. Given a supervised dataset, the parameters $\boldsymbol{\theta}$ are obtained by minimizing a loss function $\ell$ over the dataset. In natural training, a common loss choice is the cross-entropy loss, denoted as $\ell^{\mathrm{CE}}$. To compute $\ell^{\mathrm{CE}}$, we need prediction probabilities $p(\boldsymbol{x}; \boldsymbol{\theta})$, which is evaluated as $p_i(\boldsymbol{x}; \boldsymbol{\theta}) = \frac{\exp(f_i(\boldsymbol{x}; \boldsymbol{\theta}))}{\sum_j \exp(f_j(\boldsymbol{x}; \boldsymbol{\theta}))}$ element-wise.

Given a tolerance level $\epsilon$ and a distance measure norm $\mathcal{L}_p$, we say the network $f$ is robust to adversaries at a given point $\boldsymbol{x} \in \mathcal{X}$, if it satisfies

$$\arg\max_{i \in C} f_i(\boldsymbol{x}'; \boldsymbol{\theta}) = \arg\max_{i \in C} f_i(\boldsymbol{x}; \boldsymbol{\theta}), \qquad \forall \boldsymbol{x}' \text{ s.t } \|\boldsymbol{x}' - \boldsymbol{x}\|_p \leq \epsilon. \tag{1}$$

We equivalently use a ball $\mathcal{B}_\epsilon(\boldsymbol{x}) = \{\boldsymbol{x}' \mid \|\boldsymbol{x}' - \boldsymbol{x}\|_p \leq \epsilon\}$ to represent the allowed perturbation. If $f$ is robust on $\mathcal{B}_\epsilon$, we call $\mathcal{B}_\epsilon$ a robust ball.

Evaluating the true robustness of $f$ on $\mathcal{X}$ is challenging. In practice, we replace $\mathcal{X}$ with a test set and evaluate the robustness of $f$ by measuring the percentage of the test set satisfying the condition in equation (1). To check whether the condition is met, various attack strategies are applied. Commonly used attacks are PGD based. To be specific, PGD performs the following gradient step during each iteration $t + 1$,

$$\boldsymbol{x}_{t+1} = \Pi_{\mathcal{B}_\epsilon(\boldsymbol{x})}(\boldsymbol{x}_t + \xi \cdot \mathrm{sgn}(\nabla_{\boldsymbol{x}} \ell(f(\boldsymbol{x}_t; \boldsymbol{\theta})), y)), \qquad \xi < \epsilon, \tag{2}$$

and repeats the iteration several times. Here, sgn denotes the sign function. On the other hand, FGSM uses a single gradient step

$$\boldsymbol{x}' = \boldsymbol{x} + \epsilon \cdot \mathrm{sgn}(\nabla_{\boldsymbol{x}} \ell(f(\boldsymbol{x}; \boldsymbol{\theta})), y). \tag{3}$$

Finally, we introduce the following notation to facilitate the discussion. We first assume that the neural network $f$ contains ReLU activation for the sake of clarity. This implies that the neural network is piecewise linear. As a consequence, at each $\boldsymbol{x} \in \mathcal{X}$, it is easy to find a weight matrix $W^{\boldsymbol{x}} \in \mathbb{R}^{C \times N}$ and a constant vector $\boldsymbol{b}^{\boldsymbol{x}} \in \mathbb{R}^C$ such that $f(\boldsymbol{x}; \boldsymbol{\theta}) = W^{\boldsymbol{x}} \boldsymbol{x} + \boldsymbol{b}^{\boldsymbol{x}}$. To simplify the notation, we define

$$\breve{W}^{\boldsymbol{x}} = [\, W^{\boldsymbol{x}} \mid \boldsymbol{b}^{\boldsymbol{x}} \,] \in \mathbb{R}^{C \times (N+1)}, \qquad \breve{\boldsymbol{x}}^T = [\, \boldsymbol{x}^T \mid 1 \,] \in \mathbb{R}^{1 \times (N+1)}. \tag{4}$$

As a result, at each point $\boldsymbol{x}$, we have $f(\boldsymbol{x}; \boldsymbol{\theta}) = \breve{W}^{\boldsymbol{x}} \breve{\boldsymbol{x}}$. Before we delve in to the details of the algorithm ATLAS, we first introduce a new framework for designing robust training losses.

## 4 THE GENERAL FRAMEWORK

Unlike the optimization perspective taken by ADV and the regularization viewpoint adopted by TRADES, our new loss framework is motivated from a geometric standpoint. We start by briefly mentioning its closely related regularization type approaches. Generally, in such approaches, we start from a natural image $\boldsymbol{x}$. By including a loss term for $\boldsymbol{x}$, regularization type losses ensure the model gives the correct prediction $y_{\boldsymbol{x}}$ at $\boldsymbol{x}$. Assume the network makes the correct prediction at the natural image $\boldsymbol{x}$. It follows that, there exists a robust ball $\mathcal{B}_\gamma(\boldsymbol{x})$, where $\gamma \geq 0$ is the maximum radius achievable. Previous use of adversary regularization term encourages the radius $\gamma$ of the ball $\mathcal{B}_\gamma(\boldsymbol{x})$, centered at $\boldsymbol{x}$, to increase.

We take a different direction in our new loss framework. Instead of focusing on one global ball centered at $x$, we focus on local balls centered at various points $x' \in \mathcal{B}_\epsilon(x)$ and hopefully, by combining them in a regularized way, we achieve the desired robustness $\gamma \geq \epsilon$. To facilitate the illustration, we introduce our loss framework through a binary classification problem. Let there be two classes $\mathcal{C} = \{c_1, c_2\}$. For an arbitrary point $x \in \mathcal{X}$, the correct class label is $y_x = c_1$. Since we are interested in a network's performance in classifying labels, we compute logits difference as

$$f_1(x; \theta) - f_2(x; \theta) = d\breve{W}^x \breve{x} \in \mathbb{R},$$

where $d = [1, -1]$ is a row vector. For a binary classification problem with cross-entropy loss, the loss decreases monotonically with the increase of the value $d\breve{W}^x\breve{x}$. To achieve the desired robustness, we require $d\breve{W}^{x'}\breve{x}' > 0$ for all $x' \in \mathcal{B}_\epsilon(x)$. When cross entropy loss $\ell$ is considered, we need $\ell(x') < -\log(0.5)$ for all $x' \in \mathcal{B}_\epsilon(x)$. In Figure 1, we show a potential loss curve on $\mathcal{B}_\epsilon(x)$ by fixing the value of $x$ on all dimensions apart from one. Our loss framework consists of a local component and a global component. Both components update the parameters of the network model.

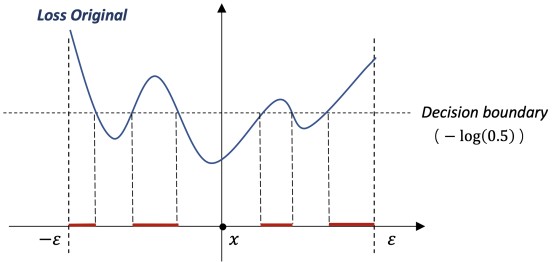

Figure 1: Original loss curve on $\mathcal{B}_\epsilon(x)$ by fixing the value of $x$ on all dimensions apart from one. Points with the loss above the decision boundary $-\log 0.5$ are adversarial points and marked as red.

### 4.1 LOCAL ROBUSTNESS

For the local component, at each given local point $x' \in \mathcal{B}_\epsilon(x)$, the goal is to attain a robust ball $\mathcal{B}_{\gamma'}(x')$ with large radius. We maximize the use of adversaries by treating them as local center points $x'$. In other words, given an adversary $x' \in \mathcal{B}_\epsilon(x)$, we need to satisfy two requirements: the model predicts the required label $y_x$ at $x'$ and the radius $\gamma'$ of $\mathcal{B}_{\gamma'}(x')$ should be enlarged. When compared with ADV, the additional second requirement allows us to achieve improved robustness performance even when weak adversaries are used.

The first requirement can be easily met by applying a standard loss term at $x'$ with the label $y_x$. For the second requirement, to push away the decision boundary, various methods including Jacobian regularizers (Jakubovitz & Giryes, 2018; Ross & Doshi-Velez, 2018), MMA (Ding et al., 2019) and MMR (Croce & Hein, 2019) have been introduced. An expected loss curve after introducing the local component is shown in Figure 2.

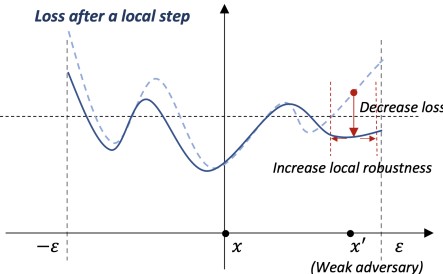

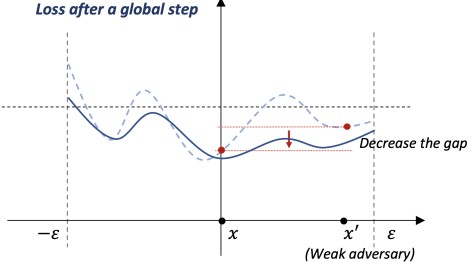

Figure 2: Loss curve after a local step. The dashed line is the original loss curve. Provided with a weak adversary $x'$, the local step aims to decrease the loss at $x'$ while increase local robustness so the local robust ball centered at $x'$ has a large radius.

Figure 3: Loss curve after a global step. The dashed line is the loss curve after a local step. To further smooth the loss curve and to remove adversaries between $x$ and $x'$, we penalize the difference $p(x; \theta) - p(x'; \theta)$. As a result, the gap between $\ell(x)$ and $\ell(x')$ is decreased.

### 4.2 GLOBAL REGULARIZATION

In terms of the global component, we combine local balls together in a controlled way to further improve robustness. We first make the following observation.

**Proposition 1.** *Consider a ball $\mathcal{B}_\epsilon(\boldsymbol{x})$ at an arbitrary point $\boldsymbol{x} \in \mathcal{X}$. For any $\boldsymbol{x}^* \in \mathcal{B}_\epsilon(\boldsymbol{x})$, we define $\breve{W}^{\boldsymbol{x}^*}$ and $\breve{\boldsymbol{x}}^*$ accordingly, as in equation (4). Let $u$ be a constant such that, for all possible $\breve{W}^{\boldsymbol{x}^*}$, the following is satisfied $\|\breve{W}^{\boldsymbol{x}^*}\|_F \leq u$. Assume $\boldsymbol{x}^1$ and $\boldsymbol{x}^2$ are arbitrary chosen points in $\mathcal{B}_\epsilon(\boldsymbol{x})$. Then for any*

$$\boldsymbol{x}_\eta = \eta \cdot \boldsymbol{x}^1 + (1 - \eta) \cdot \boldsymbol{x}^2, \tag{5}$$

*we have*

$$\boldsymbol{d}\breve{W}^{\boldsymbol{x}_\eta}\breve{\boldsymbol{x}}_\eta \geq \frac{1}{2}(\boldsymbol{d}\breve{W}^{\boldsymbol{x}^1}\breve{\boldsymbol{x}}^1 + \boldsymbol{d}\breve{W}^{\boldsymbol{x}^2}\breve{\boldsymbol{x}}^2) - u \cdot L, \tag{6}$$

*where $L = \sqrt{2}(2\|\breve{\boldsymbol{x}}_\eta\|_2 + \frac{1}{2}\|\breve{\boldsymbol{x}}^1 - \breve{\boldsymbol{x}}^2\|_2)$ is a constant.*

We now consider the combination of local robust balls. At each loss computation, we have two points: the natural image $\boldsymbol{x}$ and the adversary $\boldsymbol{x}'$. Assume the model makes the correct prediction at the natural image and, after the local component loss, at $\boldsymbol{x}'$, so we have two non-empty local robust balls $\mathcal{B}_\gamma(\boldsymbol{x})$ and $\mathcal{B}_{\gamma'}(\boldsymbol{x}')$. We further assume these two balls are disjoint for the sake of simplicity. The same underlying idea should work for more general cases. The goal is that, with the global regularization step, points that are misclassified even after the local step can be correctly predicted.

To do so, given that local robust balls $\mathcal{B}_\gamma(\boldsymbol{x})$ and $\mathcal{B}_{\gamma'}(\boldsymbol{x}')$ are disjoint, we assume there exists an adversary point $\boldsymbol{x}^* \in \mathcal{B}_\epsilon(\boldsymbol{x})$ satisfying

$$\boldsymbol{x}^* = \eta^* \cdot \boldsymbol{x} + (1 - \eta^*) \cdot \boldsymbol{x}', \tag{7}$$

where $\eta^* \in (0, 1)$ is a constant depending on $\mathcal{B}_\gamma(\boldsymbol{x})$ and $\mathcal{B}_{\gamma'}(\boldsymbol{x}')$. Then a direct application of Proposition 1 gives

$$\begin{aligned}
\boldsymbol{d}\breve{W}^{\boldsymbol{x}^*}\breve{\boldsymbol{x}}^* &\geq \frac{1}{2}(\boldsymbol{d}\breve{W}^{\boldsymbol{x}}\breve{\boldsymbol{x}} + \boldsymbol{d}\breve{W}^{\boldsymbol{x}'}\breve{\boldsymbol{x}}') - u \cdot L \\
&= \frac{1}{2}\boldsymbol{d}(\breve{W}^{\boldsymbol{x}}\breve{\boldsymbol{x}} - \breve{W}^{\boldsymbol{x}'}\breve{\boldsymbol{x}}') + \boldsymbol{d}\breve{W}^{\boldsymbol{x}'}\breve{\boldsymbol{x}}' - u \cdot L
\end{aligned} \tag{8}$$

where $u$ and $L$ are defined accordingly.

To make $\boldsymbol{x}^*$ no longer an adversary (that is, to encourage $\boldsymbol{d}\breve{W}^{\boldsymbol{x}^*}\breve{\boldsymbol{x}}^* > 0$) after global combination, we should make the lower bound on the right as high as possible. Since both $\boldsymbol{d}\breve{W}^{\boldsymbol{x}}\breve{\boldsymbol{x}}$ and $\boldsymbol{d}\breve{W}^{\boldsymbol{x}'}\breve{\boldsymbol{x}}'$ have the same max value theoretically, we can increase their sum by first increasing the value of $\boldsymbol{d}\breve{W}^{\boldsymbol{x}'}\breve{\boldsymbol{x}}'$ and then decrease the distance between $\boldsymbol{d}\breve{W}^{\boldsymbol{x}}\breve{\boldsymbol{x}}$ and $\boldsymbol{d}\breve{W}^{\boldsymbol{x}'}\breve{\boldsymbol{x}}'$. We recall that during the local robustness step, we have introduced cross-entropy loss at $\boldsymbol{x}'$, which directly maximizes the value of $\boldsymbol{d}\breve{W}^{\boldsymbol{x}'}\breve{\boldsymbol{x}}'$. In addition, enlarging local robust balls often leads to a decreased value of $u$. Given that $\boldsymbol{d}\breve{W}^{\boldsymbol{x}'}\breve{\boldsymbol{x}}'$ and $u$ are already optimized, a sound global step is to minimise the distance between $\boldsymbol{d}\breve{W}^{\boldsymbol{x}}\breve{\boldsymbol{x}}$ and $\boldsymbol{d}\breve{W}^{\boldsymbol{x}'}\breve{\boldsymbol{x}}'$. We note that, for the distance to be zero, it is sufficient to have the relative differences $f_2(\boldsymbol{x}'; \boldsymbol{\theta}) - f_1(\boldsymbol{x}'; \boldsymbol{\theta})$ and $f_2(\boldsymbol{x}; \boldsymbol{\theta}) - f_1(\boldsymbol{x}; \boldsymbol{\theta})$ to be equal rather than requiring an equality between logits $f(\boldsymbol{x}; \boldsymbol{\theta})$ and $f(\boldsymbol{x}'; \boldsymbol{\theta})$. To account for this fact, we use prediction probabilities instead. The global regularization step requires minimizing the prediction probability difference $p(\boldsymbol{x}; \boldsymbol{\theta}) - p(\boldsymbol{x}'; \boldsymbol{\theta})$. As the result, we are likely to turn $\boldsymbol{x}^*$ into a correctly predicted point after the global step, as illustrated in Figure 3.

When the loss surface is considered, the global component loss encourages the loss surface to be smoothened and flattend on $\mathcal{B}_\epsilon(\boldsymbol{x})$, so the global component is helpful even when no such adversaries $\boldsymbol{x}^*$ exist. As suggested in (Zhang et al., 2019b; Cisse et al., 2017; Ross & Doshi-Velez, 2018; Moosavi-Dezfooli et al., 2017), these properties of the loss surface are desired and are possibly indispensable for models to be robust.

Overall, various losses can be constructed under this framework. Depending on the problem at hand, one can choose an appropriate regularizer for increasing local robust balls while selecting a suitable metric for penalizing the prediction probability difference.

## 5 ATLAS

Guided by the goal of effective fast robust training, we propose a new training algorithm ATLAS, which stands for Adversarial Training via LocAl Stability. We mention that local stability, in our context, means stability of prediction, so it is the same as local robustness. ATLAS based on the framework. We introduce and explain our choices for local and global components.

### 5.1 ATLAS-LOCAL

Many existing approaches for increasing robust ball radius are computationally expensive. In this work, we adopt the standard Jacobian approach to meet our local component requirements based on the fact that an efficient estimation algorithm for Jacobian has been developed in Hoffman et al. (2019). We briefly introduce how Jacobian can be used to increase local robustness.

Given a local point $\boldsymbol{x}'$, let the correct class label be $c = y_{\boldsymbol{x}}$. Then for any $c' \neq c$, the boundary hyper-surface separating classes $c$ and $c'$ consists of points $\boldsymbol{x}^b$ satisfying

$$f_c(\boldsymbol{x}^b; \boldsymbol{\theta}) - f_{c'}(\boldsymbol{x}^b; \boldsymbol{\theta}) = 0. \tag{9}$$

Applying the standard formula for computing the distance between a point and a hyper-plane [1], we get the first order approximation distance $d_{c'}$ of $\boldsymbol{x}'$ to the boundary hyper-surface in equation (9) under the $l_2$ norm as

$$d_{c'} = \frac{|f_c(\boldsymbol{x}'; \boldsymbol{\theta}) - f_{c'}(\boldsymbol{x}'; \boldsymbol{\theta})|}{\|\nabla_{\boldsymbol{x}'} f_c(\boldsymbol{x}'; \boldsymbol{\theta}) - \nabla_{\boldsymbol{x}'} f_{c'}(\boldsymbol{x}'; \boldsymbol{\theta})\|_2}. \tag{10}$$

Since the above equation holds true for any $c'$, we conclude the model is robust on an $l_2$ norm ball centered at $\boldsymbol{x}'$ with radius $d := \min_{c' \neq c} d_{c'}$. To maximize the radius $d$, we borrow the following proposition from Jakubovitz & Giryes (2018), which introduced a Jacobian regularizer to the natural loss, to provide a lower bound for $d$.

**Proposition 2.** *Assume the model is making the correct prediction $c = y_{\boldsymbol{x}}$ for $\boldsymbol{x}'$ and the distance metric is measured via $l_2$ norm. The first order approximation of the minimum perturbation $d$ that is required to find an adversary example is lower bounded by*

$$d \geq \frac{1}{\sqrt{2} \|J(\boldsymbol{x}')\|_F} \min_{c' \neq c} |f_c(\boldsymbol{x}'; \boldsymbol{\theta}) - f_{c'}(\boldsymbol{x}'; \boldsymbol{\theta})|, \tag{11}$$

*where $J(\boldsymbol{x}') = \nabla_{\boldsymbol{x}'} f(\boldsymbol{x}'; \boldsymbol{\theta})$ is the Jacobian matrix computed at $\boldsymbol{x}'$ and $\|\cdot\|_F$ is the Frobenius norm.*

For a larger distance $d$, we need to both increase the value of $|f_c(\boldsymbol{x}'; \boldsymbol{\theta}) - f_{c'}(\boldsymbol{x}'; \boldsymbol{\theta})|$ and decrease the Frobenius norm of $J(\boldsymbol{x}')$. The first term can be easily taken care of by using a cross-entropy loss on $\boldsymbol{x}'$ while for the second term, we can include a cheap approximation of $\|J(\boldsymbol{x}')\|_F^2$, denoted by $\|J(\boldsymbol{x}')\|_F^{approx}$ as a penalty term in our loss. Using the idea of random projection, Hoffman et al. (2019) shows theoretically and empirically that $\|J(\boldsymbol{x}')\|_F^{approx}$ can be estimated with high quality by one backward pass regardless the total number of classes $C$.

To combine the above analyses, for the local robustness component, we introduce the following loss,

$$\ell_{\text{local}}(\boldsymbol{x}') = \ell^{CE}(\boldsymbol{x}') + \alpha \cdot \|J(\boldsymbol{x}')\|_F^{approx}, \tag{12}$$

where $\alpha$ is a positive scalar. It is worth noting that our local loss shown in equation (12) is different from that of (Jakubovitz & Giryes, 2018; Hoffman et al., 2019; Ross & Doshi-Velez, 2018) as the loss is evaluated at an adversary $\boldsymbol{x}'$ instead of the image $\boldsymbol{x}$. In the fast robust training setting, $\boldsymbol{x}'$ is a weak adversary obtained through one-step FGSM.

### 5.2 ATLAS-GLOBAL

To penalize the prediction probability difference, a natural and frequently used choice is Kullback–Leibler (KL) distance. We thus formulate the global loss as

$$\ell_{\text{global}}(\boldsymbol{x}') = \beta \cdot \text{KL}(p(\boldsymbol{x}'; \boldsymbol{\theta}) \| p(\boldsymbol{x}; \boldsymbol{\theta})), \tag{13}$$

where $\beta$ is a positive constant.

---

[1] in this case, the point should be $\boldsymbol{x}'$ and the hyper-plane is tangent to the boundary hyper-surface.

Although generally $\mathrm{KL}(p(\boldsymbol{x}';\boldsymbol{\theta})\|p(\boldsymbol{x};\boldsymbol{\theta})) \neq \mathrm{KL}(p(\boldsymbol{x},\boldsymbol{\theta})\|p(\boldsymbol{x}',\boldsymbol{\theta}))$ due to the asymmetric nature of KL distance, the same underlying nature of both terms means it is sufficient to use one direction for the penalty. In our loss, we have use the prediction distribution at $\boldsymbol{x}'$ to be the base distribution. There are two reasons for this choice: firstly, we want it to be consistent with the fact that the local component loss is computed on $\boldsymbol{x}'$; and secondly, we hope it could mitigate a possible over-fitting issue. If a model over-fits at the original image $\boldsymbol{x}$ causing an element $p_i(\boldsymbol{x};\boldsymbol{\theta})$ of $p(\boldsymbol{x};\boldsymbol{\theta})$ to be close to 0, the distance between $p_i(\boldsymbol{x};\boldsymbol{\theta})$ and $p_i(\boldsymbol{x}';\boldsymbol{\theta})$ would not be penalized effectively. In extreme cases, the difference will not be penalized at all if $p_i(\boldsymbol{x};\boldsymbol{\theta}) = 0$. Using $p(\boldsymbol{x}';\boldsymbol{\theta})$ as the base might alleviate this issue, as $\boldsymbol{x}'$ keeps changing during each epoch. Under the same reasoning, we point out TRADES relies on finding strong adversaries. Since it uses the natural image $\boldsymbol{x}$ to compute cross-entropy loss and as the base distribution in the KL regularizer, TRADES cannot use adversaries effectively when they are weak. This is consistent with what we see in the experiments section that is, TRADES with one step FGSM is largely outperformed by other methods on challenging datasets. Furthermore, in Appendix C.1, we show that when putting more emphasis on adversaries by replacing $\boldsymbol{x}$ with $\boldsymbol{x}'$ in TRADES, improved robust accuracy can be achieved.

## 5.3 Final Loss

Integrating the above analyses, we propose the following as the final loss for ATLAS. Given $\alpha > 0$ and $\beta > 0$, the loss is formulated as

$$\ell_{\mathrm{ATLAS}} = \frac{1}{|\mathcal{X}|} \sum_{\boldsymbol{x}\in\mathcal{X}} \underbrace{\ell^{\mathrm{CE}}(\boldsymbol{x}') + \alpha \cdot \|J(\boldsymbol{x}')\|_F^{approx}}_{\text{local}} + \underbrace{\beta \cdot \mathrm{KL}(p(\boldsymbol{x}';\boldsymbol{\theta})\|p(\boldsymbol{x};\boldsymbol{\theta}))}_{\text{global}}, \quad \boldsymbol{x}' \in \mathcal{B}_\epsilon(\boldsymbol{x}). \quad (14)$$

We mention that although our analyses are carried out in $l_2$ norm, our results are easily generalizable to other norms. To ensure the model makes correct predictions at natural images $\boldsymbol{x} \in \mathcal{X}$ and for effective patch combination, we adopt an adaptive value scheme for the pre-determined perturbation epsilon. Specifically, we start by setting $\epsilon = 0$ and gradually increases its value to the required number over epochs during the initial stage of training. Assume the final value of $\epsilon$ is $v$, to be consistent with the magnitude of $\epsilon$, we replace $\alpha$ and $\beta$ as $\frac{\epsilon}{v} \cdot \alpha$ and $\frac{\epsilon}{v} \cdot \beta$ accordingly.

## 6 Experiments

We evaluate the performance of ATLAS on three datasets: MNIST, CIFAR-10 and CIFAR-100. We consider the cases of training robustness models with weak adversaries generated by FGSM and with strong adversaries by multi-step PGD. To be consistent with the general experimental setting, we adopt $l_\infty$ norm and use the perturbation value $\epsilon = 0.3$ for MNIST and $\epsilon = 8/255$ for CIFAR-10 and CIFAR-100. MNIST is trained on a 4-layer CNN, which consists of 2 convolutional layers followed by 2 fully connected layers. CIFAR-10 and CIFAR-100 are trained on Wide-ResNet-28-8 (Zagoruyko & Komodakis, 2016). To encourage efficient robust training, instead of a fixed learning rate scheduler that decreases the learning rate at pre-specified epoch numbers, we randomly choose a subset of the data to be the validation set and then use its robust accuracy to guide the learning rate adjustment and to terminate the training process. For MNIST, 20-step PGD is applied for computing robust accuracy on the validation set while for CIFAR-10 and CIFAR-100, 10-step PGD is used. In addition, we gradually increase epsilon value from 0 to 0.3 or 8/255 for the first 15 epochs. We use the SGD optmizer for the training. More training related details can be found in Appendix B.

**Methods** We compare against ADV and TRADES. We term methods that use weak FGSM adversaries as **one-step methods**. Following the advice from Wong et al. (2020), we apply FGSM, combined with random initialization and a step size of $1.25\epsilon$ to generate adversarial examples. One-step ADV is an implementation of (Wong et al., 2020). To be consistent with their multi-step variants, we use cross-entropy loss for computing the gradient in FGSM for ADV and ATLAS while employ KL distance for TRADES. **Multi-step methods** are trained on strong adversaries generated by PGD: 20 steps for the MNIST and 10 steps for CIFAR-10 and CIFAR-100. We also include the Jacobian penalty loss (Jakubovitz & Giryes, 2018; Hoffman et al., 2019; Ross & Doshi-Velez, 2018) due to its close relation to our local loss component. We refer to it as zero-step JAC to emphasize the fact it does not require adversaries. For each method, a range of parameters are tested and those that give both high clean accuracy and high validation robust accuracy are chosen. Results for all tested parameter choices are provided in Appendix F. Due to the limited space, the ablation study on the effect of each component of ATLAS: the local part (ATLAS-l, by setting $\beta = 0$) and the global part

(ATLAS-g, by setting $\alpha = 0$) is left to Appendix C and the same is for a detailed comparison between TRADES and ATLAS-g. In addition, we have include in Appendix D.1 an experiment to demonstrate ATLAS's effective use of weak adversaries. We compare ATLAS against a loss which always treats the natural image $\boldsymbol{x}$, instead of a weak adversary $\boldsymbol{x}'$, as the center point for the local component. In the words, the loss takes the form of combining TRADES and zero-step JAC.

**Robustness Evaluation** Extensive robustness evaluations are conducted in order to avoid a false sense of security. Black box attack generates adversaries on surrogate models with 1000 PGD steps. In terms of white box attacks, apart from an initial robustness evaluation with PGD-20, we further introduce three strong white attacks: PGD-1000, Untargeted (Un-T) attack (Carlini & Wagner, 2017) and Multi-targeted (Multi-T) attack (Gowal et al., 2019). Untargeted attack uses the loss $f_u(\boldsymbol{x}; \boldsymbol{\theta}) - f_c(\boldsymbol{x}; \boldsymbol{\theta})$, where $c$ is the correct class and $u = \arg\max_{i \neq c} f_i(\boldsymbol{x})$. We allow $u$ to change during each gradient step. Regarding Multi-targeted attack, we perform attack with the loss $f_i(\boldsymbol{x}; \boldsymbol{\theta}) - f_c(\boldsymbol{x}; \boldsymbol{\theta})$ for all $i \in \mathcal{C}$ such that $i \neq c$. On the big Wide-ResNet model, we run 10 restarts with 100 steps for Untargeted attack and 5 restarts with 20 steps for Multi-target attack. We reduce the number of restarts and steps for the Multi-target attack to account for the fact that each incorrect class needs to be tested. Due to the large number of classes, Multi-target is not performed on CIFAR-100. On the other hand, we perform 20 restarts with 50 steps for both attacks on the small CNN model. For each method, attack results are reported for a model at the epoch that gives the best validation robust accuracy during the training. On the challenging datasets CIFAR-10 and CIFAR-100, two more robust evaluations are carried out to obtain a more comprehensive understanding of models' robustness performance. Specifically, we apply an hard-label attack, RayS (Chen & Gu, 2020) and an ensemble of diverse parameter-free attacks, AutoAttack (Croce & Hein, 2020). Models' robustness performance under these two evaluations are reported in Appendix D.2.

We summarize attack results in Table 1. We mention that with a learning rate scheduler, which is guided by the robust accuracy on the validation set, all models achieved their best performance around 30 epochs, except 12 epochs for zero-step JAC on MNIST. In terms of robustness performance, slight performance improvements for all methods and narrower performance gaps will be observed if all models are trained for 70 epochs with a fixed learning rate scheduler on MNIST. On the other hand, on CIFAR-10 and CIFAR-100, we found all methods perform better with the guided learning rate scheduler. Since we are interested in efficient robust learning, we use the guided learning rate scheduler for all experiments.

**Evaluations on CIFAR-10** We study the challenging CIFAR-10 first. For zero-step Jac, we observe that to achieve a roughly $30\%$ robust accuracy, nominal accuracy is dropped to below $65\%$. Directly including a Jacobian penalty at natural images could thus lead to over-regularization issues. This may also be the reason why the method is used as a post-processing technique in (Jakubovitz & Giryes, 2018). On the other hand, by using local adversaries instead of natural images, ATLAS-l manages to obtain robust accuracy without sacrificing the nominal accuracy too much, which is demonstrated in the Appendix. When TRADES is considered, we find that its effectiveness relies on strong adversaries: robust accuracy increases sharply from one-step case to multi-step case. Since TRADES computes loss at natural images and use them as the base distribution in the KL penalty, weak adversaries are not effectively used.

In addition, we constantly find for one-step TRADES that, after a certain number of epochs, the percentage of the model making incorrect decisions on weak adversaries falls sharply, leading to a sudden drop of more than $10\%$ on validation robust accuracy. This phenomena is referred to as catastrophic overfitting (Wong et al., 2020; Andriushchenko & Flammarion, 2020). We observe that when this behaviour occurs, the approximated value $\|J(\boldsymbol{x}')\|_F^{approx}$ increases steeply, which is consistent with what has been observed in Andriushchenko & Flammarion (2020). When dealing with catastrophic overfitting is the main concern, Andriushchenko & Flammarion (2020) argue the key is to increase local linearity and they introduce a regularizer to maximize gradient alignment for points within the perturbation ball. In our case, the issue could be similarly resolved by increasing the coefficient $\alpha$ to encourage local linearity. However, since the goal of this project is to achieve high robust accuracy fast, we do not restrict ourselves to models without catastrophic overfitting only, which are likely to compensate robust accuracy for stability. Instead, we use early termination and consider a wider range of models. We mention that the same phenomenon is observed on one-step ADV and on one-step ATLAS when $\alpha$ is small but less frequently. More discussions on catastrophic overfitting can be found in Appendix E.

Table 1: Robustness performance for MNIST on a small CNN and for CIFAR-10 and CIFAR-100 on Wide-Resnet 28-8. The higher the better.

| | Model | Clean accuracy | Black box attack | White box attack | | | | Time per batch |
|---|---|---|---|---|---|---|---|---|
| | | | | PGD-20 attack | PGD-1000 attack | Un-T attack | Multi-T attack | |
| MNIST | zero-step JAC | 98.34% | 89.84% | 58.09% | 2.48% | 28.83% | 28.72% | 0.012s |
| | one-step ADV | **99.49%** | 96.26 % | 96.29% | 83.25% | 91.02% | 90.62% | 0.009s |
| | one-step TRADES | 99.40% | 96.21% | 96.28% | 83.05% | 90.66% | 90.02% | 0.016s |
| | one-step ATLAS | 99.47% | **96.36%** | **96.84%** | **89.56%** | **92.44%** | **92.04%** | 0.018s |
| | twenty-step ADV | 99.48% | 96.40% | 97.16% | **92.32%** | 93.01% | **92.87%** | 0.077s |
| | twenty-step TRADES | **99.49%** | 96.21% | 97.01% | 90.51% | 92.49% | 92.21% | 0.097s |
| | twenty-step ATLAS | 99.44% | **96.45%** | **97.19%** | 92.20% | **93.03%** | 92.74% | 0.086s |
| CIFAR-10 | zero-step JAC | 63.24% | 60.45% | 34.40% | 34.34% | 31.13% | 30.32% | 0.47s |
| | one-step ADV | 86.24% | 82.53% | 45.73% | 44.98% | 45.14% | 42.97% | 0.25s |
| | one-step TRADES | **86.84%** | **82.63%** | 40.03% | 39.41% | 39.31% | 38.14% | 0.55s |
| | one-step ATLAS | 84.52% | 81.06% | **49.01%** | **48.59%** | **48.54%** | **46.55%** | 0.73s |
| | ten-step ADV | **84.77%** | 82.53% | 50.60% | 50.22% | 49.74% | 47.96% | 1.38s |
| | ten-step TRADES | 84.01% | **82.63%** | 52.70% | 52.45% | 50.29% | 49.66% | 2.02s |
| | ten-step ATLAS | 82.94% | 81.06% | **53.45%** | **53.10%** | **51.51%** | **50.16%** | 1.88s |
| CIFAR-100 | zero-step JAC | 47.96% | 44.67% | 15.39% | 15.18% | 14.35% | - | 0.47s |
| | one-step ADV | 48.88% | 45.59% | 19.89% | 19.72% | 19.04% | - | 0.25s |
| | one-step TRADES | **62.58%** | **57.64%** | 16.93% | 16.21% | 14.83% | - | 0.53s |
| | one-step ATLAS | 60.98% | 57.04% | **26.30%** | **25.94%** | **26.28%** | - | 0.73s |
| | ten-step ADV | **60.72%** | **58.15%** | 27.71% | 27.43% | 26.83% | - | 1.37s |
| | ten-step TRADES | 59.76% | 57.78% | 26.27% | 26.10% | 23.86% | - | 2.00s |
| | ten-step ATLAS | 59.62% | 57.83% | **29.12%** | **28.89%** | **27.77%** | - | 1.87s |

Overall, one-step ATLAS outperforms other one-step methods in all strong white box attacks. Even when compared with methods trained with 10 steps PGD, the performance gap between one-step ATLAS and multi-step ADV under the strongest Multi-T attack is below $1.5\%$. Although adding local and global penalties result in extra computational time for one-step ATLAS (0.73s) than one-step ADV (0.25s), the fact that ATLAS achieves comparable high robust accuracy to that of multi-step ADV (1.38s) with weak adversaries still allows a roughly $50\%$ speed-up in training. For a fair comparison, we also evaluate an ADV model with 5 PGD steps. The 5-step ADV models requires slightly more time (0.75s per batch) but is less robust than one-step ATLAS by reaching $44.93\%$ accuracy under the Multi-T attack. When trained on strong adversaries, multi-step ATLAS wins over the other two multi-step methods in all strong white box attacks.

**Evaluations on CIFAR-100 and MNIST** Similar performances are observed on the CIFAR-100. The key fact to notice here is that the per batch training cost required for CIFAR-100 is similar to that of CIFAR-10. Since we apply an efficient approximation to evaluate $\|J(x)\|_F^2$, the increase of total class number does not result in a rise of total computational cost. Again, we evaluate ADV models with 5 PGD steps, which requires 0.75s per batch and reaches $25.21\%$ robust accuracy under Un-T attack. On MNIST, one-step ATLAS performs the best as well. Since MNIST is a simple dataset, all multi-step methods obtain high robust accuracy on the small CNN model. For the same reason, the unstable sudden drop of validation robust accuracy is not observed on any one-step method.

## 7 DISCUSSION

We have adopted a different perspective and proposed a new framework for constructing losses that allow more effective use of adversaries. Specifically, based on the framework, we introduce a novel algorithm ATLAS for fast robust training. ATLAS can be used as an initialisation technique for other complicated tasks. For instance, model trained with ATLAS can be employed as the starting point for layer-wise adversarial training in improving certified robustness. Apart from fast robust training, we believe other losses could be constructed from the framework to accommodate different problems. We leave the explorations of potential applications to various robust training setting to future research.

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

## A. PROOF FOR PROPOSITION 2

**Proposition 1.** *Consider a ball $\mathcal{B}_\epsilon(\boldsymbol{x})$ at an arbitrary point $\boldsymbol{x} \in \mathcal{X}$. For any $\boldsymbol{x}^* \in \mathcal{B}_\epsilon(\boldsymbol{x})$, we define $\breve{W}^{\boldsymbol{x}^*}$ and $\breve{\boldsymbol{x}}^*$ accordingly, as in equation (4). Let $u$ be a constant such that, for all possible $\breve{W}^{\boldsymbol{x}^*}$, the following is satisfied $\|\breve{W}^{\boldsymbol{x}^*}\|_F \leq u$. Assume $\boldsymbol{x}^1$ and $\boldsymbol{x}^2$ are arbitrary chosen points in $\mathcal{B}_\epsilon(\boldsymbol{x})$. Then for any*

$$\boldsymbol{x}_\eta = \eta \cdot \boldsymbol{x}^1 + (1 - \eta) \cdot \boldsymbol{x}^2, \tag{5}$$

*we have*

$$\boldsymbol{d}\breve{W}^{\boldsymbol{x}_\eta}\breve{\boldsymbol{x}}_\eta \geq \frac{1}{2}(\boldsymbol{d}\breve{W}^{\boldsymbol{x}^1}\breve{\boldsymbol{x}}^1 + \boldsymbol{d}\breve{W}^{\boldsymbol{x}^2}\breve{\boldsymbol{x}}^2) - u \cdot L, \tag{6}$$

*where $L = \sqrt{2}(2\|\breve{\boldsymbol{x}}_\eta\|_2 + \frac{1}{2}\|\breve{\boldsymbol{x}}^1 - \breve{\boldsymbol{x}}^2\|_2)$ is a constant.*

*Proof.* The proof is straightforward. To simplify the notation, we define $s^1 := \boldsymbol{d}\breve{W}^{\boldsymbol{x}^1}\breve{\boldsymbol{x}}^1$, $s^2 := \boldsymbol{d}\breve{W}^{\boldsymbol{x}^2}\breve{\boldsymbol{x}}^2$ and $s_\eta := \boldsymbol{d}\breve{W}^{\boldsymbol{x}_\eta}\breve{\boldsymbol{x}}_\eta$. Also, let $l$ be the lower bound $\boldsymbol{d}\breve{W}^{\boldsymbol{x}^*}\breve{\boldsymbol{x}}^* \geq l$ for any $\boldsymbol{x}^* \in \mathcal{B}_\epsilon(\boldsymbol{x})$. Such $l$ exits because $\mathcal{B}_\epsilon(\boldsymbol{x})$ is a closed and bounded ball and $f$ is continuous. We first note that

$$|s^1 - l| - |s^1 - s_\eta| \leq |s_\eta - l|.$$

Since $s^1 - l > 0$ and $s_\eta - l \geq 0$, we can remove the absolute sign and have

$$s^1 - |s^1 - s_\eta| \leq s_\eta. \tag{15}$$

The term $|s^1 - s_\eta|$ can be upper bounded as

$$\begin{aligned}
|s^1 - s_\eta| &= |\boldsymbol{d}\breve{W}^{\boldsymbol{x}^1}\breve{\boldsymbol{x}}^1 - \boldsymbol{d}\breve{W}^{\boldsymbol{x}_\eta}\breve{\boldsymbol{x}}_\eta| \\
&\leq |(\boldsymbol{d}\breve{W}^{\boldsymbol{x}^1} - \boldsymbol{d}\breve{W}^{\boldsymbol{x}_\eta})\breve{\boldsymbol{x}}_\eta| + |\boldsymbol{d}\breve{W}^{\boldsymbol{x}^1}(\breve{\boldsymbol{x}}^1 - \breve{\boldsymbol{x}}_\eta)| \\
&\leq 2u\|\boldsymbol{d}\|_2\|\breve{\boldsymbol{x}}_\eta\|_2 + u(1 - \eta)\|\boldsymbol{d}\|_2\|\breve{\boldsymbol{x}}^1 - \breve{\boldsymbol{x}}^2\|_2.
\end{aligned} \tag{16}$$

We further denote $m = 2u\|\boldsymbol{d}\|_2\|\breve{\boldsymbol{x}}_\eta\|_2$ and $v = u\|\boldsymbol{d}\|_2\|\breve{\boldsymbol{x}}^1 - \breve{\boldsymbol{x}}^2\|_2$. By replace $|s^1 - s_\eta|$ with its upper bounds equation (16), it follows that

$$s^1 - m - (1 - \eta)v \leq s_\eta. \tag{17}$$

Similarly for $s^2$, we have

$$s^2 - m - \eta v \leq s_\eta. \tag{18}$$

Finally, combining equation (17) and equation (18), we get

$$\frac{1}{2}(s^1 + s^2) - m - \frac{1}{2}v \leq s_\eta, \tag{19}$$

the desired inequality. $\qquad\square$

## B. EXPERIMENTAL DETAILS

We give more details about training and attacks in the following.

**Training** We rely on the robust accuracy on a validation dataset to guide the training process. To be more specific, we first randomly select 5000 images from the training dataset to form a validation set before training starts. During training, after each epoch, we run a multi-step PGD attack to evaluate the model's robust accuracy on the validation set. If the validation robust accuracy does not improve for a fixed number of consecutive epochs (we refer to the number as plateau epoch number), we decrease the learning rate by five. If the robust accuracy does not improve for ten consecutive epochs, we terminate the training process.

**MNIST** We use SGD optimizer with a starting learning rate of 0.01 on MNIST. Batch size is set to be 128. After each epoch, we apply a 20-step PGD attack with step size 0.01 to determine validation robust accuracy. Similarly, when generating strong adversaries for multi-step methods, 20-step PGD attack with step size 0.01 is used. For numbers reported in Table 1, zero-step JAC is trained with $\alpha_{\text{JAC}} = 0.5$; one-step and multi-step TRADES are trained with $\beta_{\text{TRADES}} = 1$; one-step ATLAS is

trained with $\alpha = 1e - 05$ and $\beta = 0.3$ while multi-step ATLAS is trained with $\alpha = 5e - 06$ and $\beta = 0.2$.

**CIFAR-10 and CIFAR-100** On both CIFAR-10 and CIFAR-100, we employ the SGD optimizer with a starting learning rate of 0.1, a momentum of 0.9 and a weight decay of $2e - 4$. A batch size of 64 is used. In terms of computing validation robust accuracy and generating strong adversaries in multi-step cases, we apply 10-step PGD attack with step size 0.007. On the CIFAR-10 dataset, the following parameters are used for the numbers reported in Table 1: zero-step JAC is trained with $\alpha_{\text{JAC}} = 0.5$; one-step TRADES is trained with $\beta_{\text{TRADES}} = 2$; multi-step TRADES is trained with $\beta_{\text{TRADES}} = 4$; one-step ATLAS is trained with $\alpha = 0.0002$ and $\beta = 5$ and multi-step ATLAS is trained with $\alpha = 0.0001$ and $\beta = 5.0$. On the CIFAR-10 dataset, we used $\alpha_{\text{JAC}} = 0.5$ for zero-step JAC; $\beta_{\text{TRADES}} = 2$ for one-step TRADES; $\beta_{\text{TRADES}} = 4$ for multi-step TRADES; $\alpha = 0.0002$ and $\beta = 5.0$ for one-step ATLAS and $\alpha = 0.0001$ and $\beta = 10.0$ for multi-step ATLAS.

**Attacks** Random initialization is applied in all attacks. Untargeted and Multi-targeted attacks are implemented by following the descriptions given in (Qin et al., 2019). For these two attacks, we consider an attack is successful if an adversary is found after a gradient update at any point during the optimization procedure.

### B.1. PLATEAU EPOCH NUMBER AND FGSM STEP-SIZE

Due to the simplicity of the MNIST dataset and similar behaviour on both CIFAR-10 and CIFAR-100, we determine the plateau epoch number and the FGSM step-size by experimenting with one-step ADV on CIFAR-10.

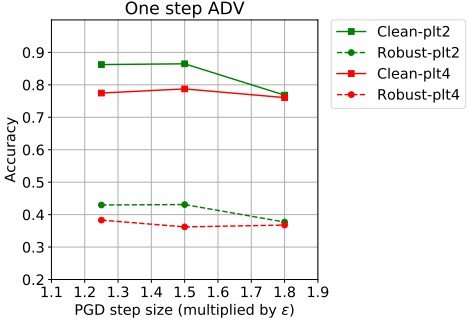

Figure 4: Clean and robust accuracy for one-step ADV at different step sizes

We use strongest Multi-T attack for evaluating model's robust accuracy. In Figure 4, the green lines show the results for plateau epochs to be 2 (plt2) while red lines are for plateau epochs to be 4 (plt4). In terms of step sizes, we tested 3 possible values: $1.25\epsilon$ (the suggested value by Wong et al. (2020)), $1.5\epsilon$ and $1.8\epsilon$. It is clear to see that with plt2, models perform better in both nominal accuracy and robust accuracy at all three step sizes. We thus use plateau epoch number 2 to adjust the learning rate in all our experiments for comparable results. When step size is considered, both $1.25\epsilon$ and $1.5\epsilon$ give satisfactory results. Since other methods show more stable performances with the step size $1.25\epsilon$, we use it for generating weak adversaries in our experiments.

## C ATLAS: ABLATION STUDIES

We perform ablation studies for ATLAS. We consider the one-step setting. Recall that ATLAS consists of two components: a local component and a global component. We test the effect of the local component by setting $\beta = 0$ to get **ATLAS-l**:

$$\ell_{\text{ATLAS-l}} = \frac{1}{|\mathcal{X}|} \sum_{\boldsymbol{x} \in \mathcal{X}} \ell^{\text{CE}}(\boldsymbol{x}') + \alpha \cdot \mathcal{A}(J(\boldsymbol{x}')), \quad \boldsymbol{x}' \in \mathcal{B}_\epsilon(\boldsymbol{x}). \tag{20}$$

and the global component by setting $\alpha = 0$ to get **ATLAS-g**:

$$\ell_{\text{ATLAS-g}} = \frac{1}{|\mathcal{X}|} \sum_{\boldsymbol{x} \in \mathcal{X}} \ell^{\text{CE}}(\boldsymbol{x}') + \beta \cdot \text{KL}(p(\boldsymbol{x}'; \boldsymbol{\theta}) \| p(\boldsymbol{x}; \boldsymbol{\theta})), \quad \boldsymbol{x}' \in \mathcal{B}_{\epsilon}(\boldsymbol{x}). \tag{21}$$

We apply the strongest Multi-T attack on CIFAR-10 and Un-T attack on CIFAR-100. Results are summarised in Table 2. When CIFAR-10 is considered, both ATLAS-l and ATLAS-g alone are effective in improving the model's robust accuracy. Combining the local and global components (one-step ATLAS) lead to a further robustness improvement. In terms of time per batch, apart from the cross-entropy loss term at the adversary in both ATLAS-l and ATLAS-g, ATLAS-l requires computing the gradient of a Jacobian approximation term and is computationally more expensive than ATLAS-g, which calculates the gradient of a KL penalty term instead. Furthermore, the fact that ATLAS-g is computationally cheaper than TRADES is because it uses cross-entropy loss to compute gradients for FGSM while TRADES uses KL distance. Similar performance is observed on CIFAR-100.

Table 2: Robustness performance for CIFAR-10 and CIFAR-100 on Wide-Resnet 28-8. The higher the better.

| | Model | Clean accuracy | Un-T attack | Multi-T attack | Time per batch |
|---|---|---|---|---|---|
| **CIFAR-10** | one-step ADV | 86.24% | - | 42.97% | 0.25s |
| | one-step TRADES | **86.84%** | - | 38.14% | 0.55s |
| | one-step ATLAS-l | 85.24% | - | 44.46% | 0.60s |
| | one-step ATLAS-g | 82.07% | - | 44.43% | 0.38s |
| | one-step ATLAS | 84.52% | - | **46.55%** | 0.73s |
| **CIFAR-100** | one-step ADV | 48.88% | 19.04% | - | 0.25s |
| | one-step TRADES | **62.58%** | 14.83% | - | 0.53s |
| | one-step ATLAS-l | 58.46% | 24.73% | - | 0.60s |
| | one-step ATLAS-g | 51.88% | 19.71% | - | 0.37s |
| | one-step ATLAS | 60.98% | **26.28%** | - | 0.73s |

We show model's performance when trained with ATLAS-l and ATLAS-g at different parameter values. Results for CIFAR-10 are summarised in Figure 5 and 6 while results for CIFAR-100 are summarised in Figure 7 and 8. For CIFAR-10, we see that the clean accuracy decreases with the increase of $\alpha$ value for ATLAS-l but robust accuracy retains at the similar level. In terms of ATLAS-g, the same trend is observed and when the value of $\beta$ is large, both clean and robust accuracy decline. On CIFAR-100, there is a big drop in robust accuracy for ATLAS-l when $\alpha$ rises while robust accuracy for ATLAS-g is relatively insensitive to the change of $\beta$ value.

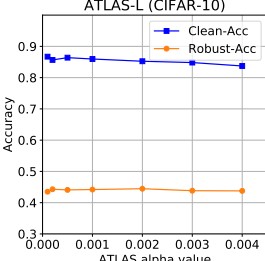

Figure 5: Clean and robust accuracy for ATLAS-l at different $\alpha$ on CIFAR-10

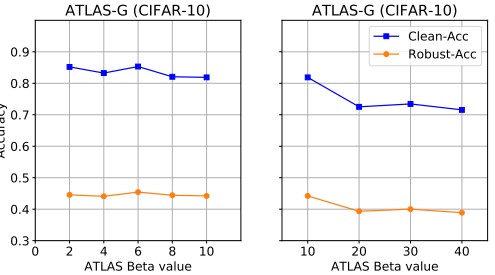

Figure 6: Clean and robust accuracy for ATLAS-l at different $\beta$ on CIFAR-10

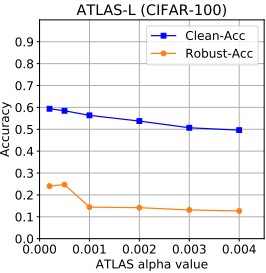
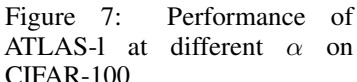
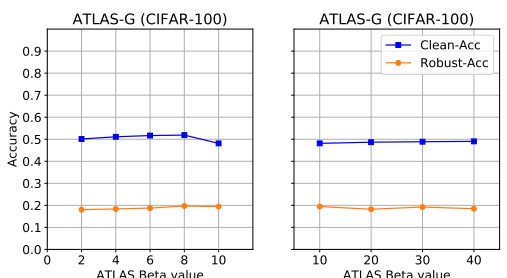

Figure 7: Performance of ATLAS-l at different $\alpha$ on CIFAR-100

Figure 8: Clean and robust accuracy for ATLAS-l at different $\beta$ on CIFAR-100

## C.1 ATLAS-G VS. TRADES

ATLAS-g and TRADES take similar forms. The only differences between these two methods are: firstly, ATLAS-g computes cross-entropy loss at the adversary while TRADES at the natural image; secondly, ATLAS-g uses cross-entropy loss to find a gradient in FGSM while TRADES employs KL distance.

On CIFAR-10, it is clear that ATLAS-g outperforms TRADES on both clean and robust accuracy for all tested $\beta$ values. This observation supports the fact that ATLAS-g, by computing the loss at weak adversaries, allows a more effective use of them. Maximizing the use of weak adversaries is important in fast robust training setting. In terms of CIFAR-100, ATLAS-g outperforms TRADES on robust accuracy. However, TRADES achieves higher clean accuracy on small $\beta$ values. The regularization effect of ATLAS-g could be higher than TRADES.

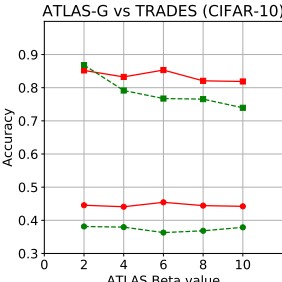
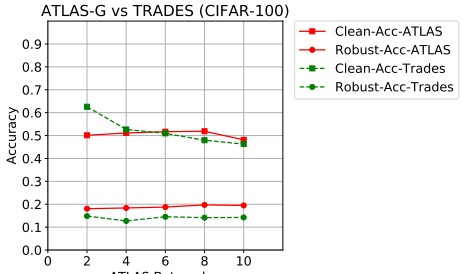

Figure 9: ATLAS-g vs. TRADES on CIFAR-10

Figure 10: ATLAS-g vs. TRADES on CIFAR-100

## D ADDITIONAL EXPERIMENTS

We compare ATLAS against another loss of similar form. Furthermore, we perform two extra attacks to obtain more comprehensive evaluations of all tested models. Due to the simplicity of MNIST, we focus on the challenging datasets CIFAR-10 and CIFAR-100 only.

## D.1 ADDITIONAL LOSS

We study an additional loss to demonstrate ATLAS's effective use of weak adversaries. We recall that within our framework, the key to maximize the use of weak adversaries is to treat them as center points of local balls in the local component. We thus consider the following loss

$$\ell_{\text{TradesJac}} = \frac{1}{|\mathcal{X}|} \sum_{\boldsymbol{x} \in \mathcal{X}} \underbrace{\ell^{\text{CE}}(\boldsymbol{x}) + \alpha_{\text{tj}} \cdot \|J(\boldsymbol{x})\|_F^{approx}}_{\text{local}} + \underbrace{\beta_{\text{tj}} \cdot \text{KL}(p(\boldsymbol{x};\boldsymbol{\theta}) \| p(\boldsymbol{x}';\boldsymbol{\theta}))}_{\text{global}}, \quad \boldsymbol{x}' \in \mathcal{B}_\epsilon(\boldsymbol{x}).$$

We term the loss as TradesJac, as it can also been seen as a combination of the TRADES loss and the zero-step JAC loss. Adversaries are computed by using the gradient of the KL-distance. The main difference between TradesJac and ATLAS is in the local component, where TradesJac uses natural images as center points for local balls instead of weak adversaries.

We train models using TradesJac with various sets of values for $\alpha_{tj}$ and $\beta_{tj}$ and reported results for the model with the best validation robust accuracy. Results can be found in Table 3. On both CIFAR-10 and CIFAR-100, we have set $\alpha_{tj} = 0.0002$ and $\beta_{tj} = 0.5$. We have also include other one-step methods for easy comparison. It is clear to see that ATLAS outperforms TradesJac in all attacks, confirming that ATLAS is able to employ weak adversaries more effectively in achieving robust accuracy.

Table 3: Robustness performance for one-step methods on CIFAR-10 and CIFAR-100. All models are trained with Wide-Resnet 28-8. The higher the better.

| | Model | Clean accuracy | Black box attack | White box attack | | | | Time per batch |
| | | | | PGD-20 attack | PGD-1000 attack | Un-T attack | Multi-T attack | |
|---|---|---|---|---|---|---|---|---|
| CIFAR-10 | one-step ADV | 86.24% | 82.53% | 45.73% | 44.98% | 45.14% | 42.97% | 0.25s |
| | one-step TRADES | **86.84%** | **82.63%** | 40.03% | 39.41% | 39.31% | 38.14% | 0.55s |
| | one-step ATLAS | 84.52% | 81.06% | **49.01%** | **48.59%** | **48.54%** | **46.55%** | 0.73s |
| | one-step TradesJac | 84.38% | 81.54% | 46.04% | 45.75% | 44.60% | 43.59% | 1.14s |
| CIFAR-100 | one-step ADV | 48.88% | 45.59% | 19.89% | 19.72% | 19.04% | - | 0.25s |
| | one-step TRADES | **62.58%** | **57.64%** | 16.93% | 16.21% | 14.83% | - | 0.53s |
| | one-step ATLAS | 60.98% | 57.04% | **26.30%** | **25.94%** | **26.28%** | - | 0.73s |
| | one-step TradesJac | 58.15% | 54.56% | 21.22% | 20.89% | 19.78% | - | 1.15s |

## D.2 ADDITIONAL ATTACKS

We perform two additional robustness evaluations on CIFAR-10 and CIFAR-100. The first one is a hard-label attack called RayS (Chen & Gu, 2020). For both datasets, we run RayS with 40000 queries on 1000 randomly selected images from the test set. The second one is an ensemble of parameter-free attacks, named as AutoAttack (Croce & Hein, 2020). AutoAttack consists of two variants of PGD attack, an attack focused on gradient-masking and a black-box attack. We have used the standard version of AutoAttack for all evaluations. Results for CIFAR-10 and CIFAR-100 can be found in Table 4 and Table 5 respectively. ATLAS gives the most robust performance among all methods in both weak and strong adversaries cases.

Table 4: Robustness performance for CIFAR-10 on Wide-Resnet 28-8. The higher the better.

| | Model | RayS attack | Auto attack |
|---|---|---|---|
| | zero-step JAC | 33.50% | 30.03% |
| CIFAR-10 | one-step ADV | 50.20% | 42.49% |
| | one-step TRADES | 45.10% | 37.74% |
| | one-step ATLAS | **52.70%** | **46.16%** |
| | one-step TradesJac | 50.20% | 43.16% |
| | ten-step ADV | **54.40%** | 47.56% |
| | ten-step TRADES | 54.00% | 49.40% |
| | ten-step ATLAS | 53.20% | **49.82%** |

Table 5: Robustness performance for CIFAR-100 on Wide-Resnet 28-8. The higher the better.

| | Model | RayS attack | Auto attack |
|---|---|---|---|
| | zero-step JAC | 16.70% | 11.85% |
| CIFAR-100 | one-step ADV | 19.90% | 17.17% |
| | one-step TRADES | 19.50% | 12.75% |
| | one-step ATLAS | **28.60%** | **23.42%** |
| | one-step TradesJac | 22.60% | 17.99% |
| | ten-step ADV | 29.80% | 24.64% |
| | ten-step TRADES | 28.20% | 22.78% |
| | ten-step ATLAS | **30.60%** | **25.77%** |

# E CATASTROPHIC OVERFITTING

We give more details on catastrophic overfitting behaviour in our experiments. We first show typical training plots for both cases. The upper row of Figure 11 contains plots for a model when catastrophic overfitting is avoided and the bottom row show plots for a model when catastrophic overfitting happens. It is clear to see that catastrophic overfitting is highly associated with a sharp increase of Jacobian value, which is consistent with the observations made in (Andriushchenko & Flammarion, 2020). In our experiments with roughly 30 epochs, we find that catastrophic overfitting always occurs for one-step TRADES and ATLAS-g but less frequently for one-step ADV and one-step ATLAS. For ATLAS-l, the catastrophic overfitting can be effectively avoided by increasing the value for $\alpha$ as it directly penalizes large Jacobian value. The same strategy can be applied for the full ATLAS method. We find that, for a stable training performance, the choice of $\alpha$ depends on the value of $\beta$. That is a large $\beta$ value requires a large $\alpha$ value.

Finally, we mention that since the goal is to achieve high robust accuracy, we use early termination and consider a wider range of models regardless whether they experience catastrophic overfitting in the end.

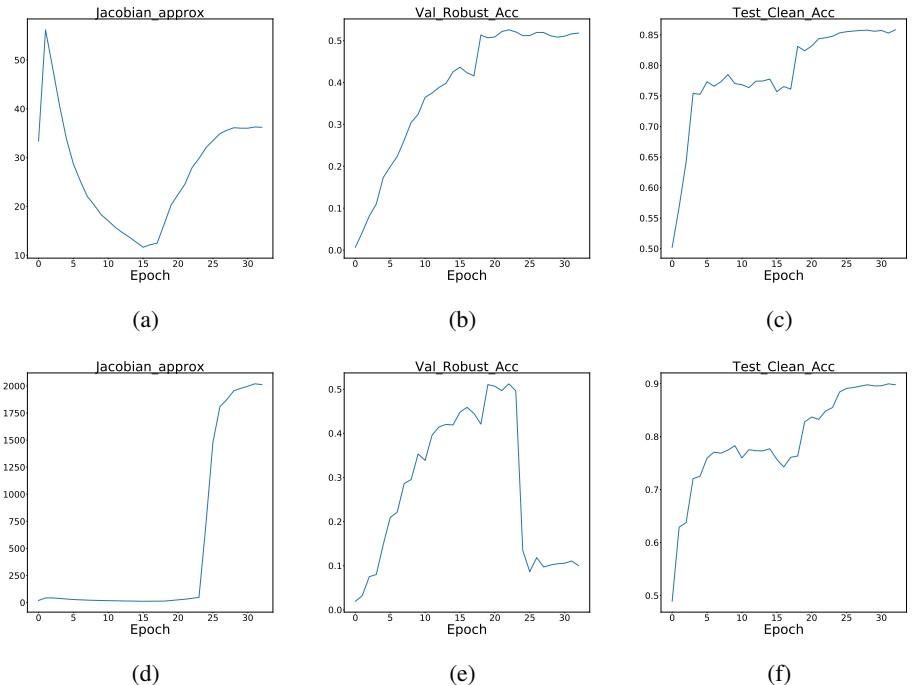

(a) (b) (c)

(d) (e) (f)

Figure 11: We show typical training plots for a model on CIFAR-10 when catastrophic overfitting is avoided in the upper row and when catastrophic overfitting happens in the bottom row. Upper row plots are for the ATLAS model with $\alpha = 0.0002$ and $\beta = 5.0$ while the bottom row plots are for the ATLAS model with $\alpha = 0$ and $\beta = 8$.. The figures on the left shows the value of $\|J(x')\|_F^{\text{approx}}$ throughout the training. Catastrophic overfitting is associated with a steep increase of the Jacobian value. The figures in the middle gives the trend for robust accuracy on the validation dataset. Validation robust accuracy drops sharply with the sudden increase of the Jacobian value. Finally, on the right, we have plots for clean accuracy on the test set after each epoch. A further increase to almost $90\%$ clean accuracy is experienced in the catastrophic overfitting case.

# F PARAMETER CHOICES: ONE-STEP CASE

We show clean accuracy and robust accuracy for each method at various parameter choices.

## F.1 Zero-Step Jac

Although zero-step Jac does not require adversaries, we used 2 epochs for adjusting learning rate via validation robust accuracy to be consistent. Model's performance over a range of $\alpha_{\text{Jac}}$ values is shown in Figure 12 for CIFAR-10 and 13 for CIFAR-100. There is a large trade-off between clean and robust accuracy for small $\alpha_{\text{Jac}}$ values and then both clean and robust accuracy decrease with the increase of $\alpha_{\text{Jac}}$.

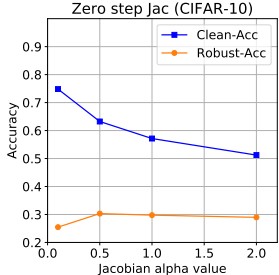

Figure 12: Clean and robust accuracy for zero-step JAC at different $\alpha_{\text{Jac}}$ on CIFAR-10

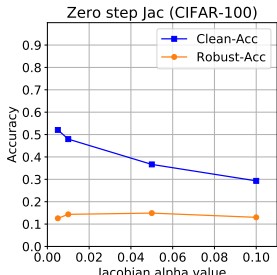

Figure 13: Clean and robust accuracy for zero-step JAC at different $\alpha_{\text{Jac}}$ on CIFAR-100

## F.2 One-Step TRADES

In Figure 14 (CIFAR-10) and 15 (CIFAR-100), we show one-step TRADES at various $\beta_{\text{TRADES}}$ values. It is easy to see that increasing the value of $\beta_{\text{TRADES}}$ mainly hurts the clean accuracy without improving robust accuracy.

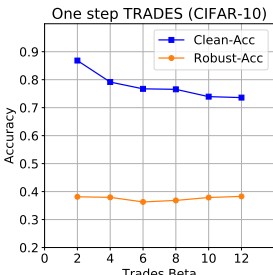

Figure 14: Clean and robust accuracy for one-step TRADES at different $\beta_{\text{TRADES}}$ on CIFAR-10

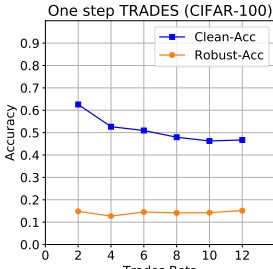

Figure 15: Clean and robust accuracy for one-step TRADES at different $\beta_{\text{TRADES}}$ on CIFAR-100

## F.3 One-Step ATLAS

In Figure 16, we give clean and robust accuracy for models trained at different $\alpha$ and $\beta$ values on CIFAR-10. In Figure 17, we give clean and robust accuracy for models trained at different $\alpha$ and $\beta$ values on CIFAR-100. A slight trade-off between clean and robust accuracy can be observed.

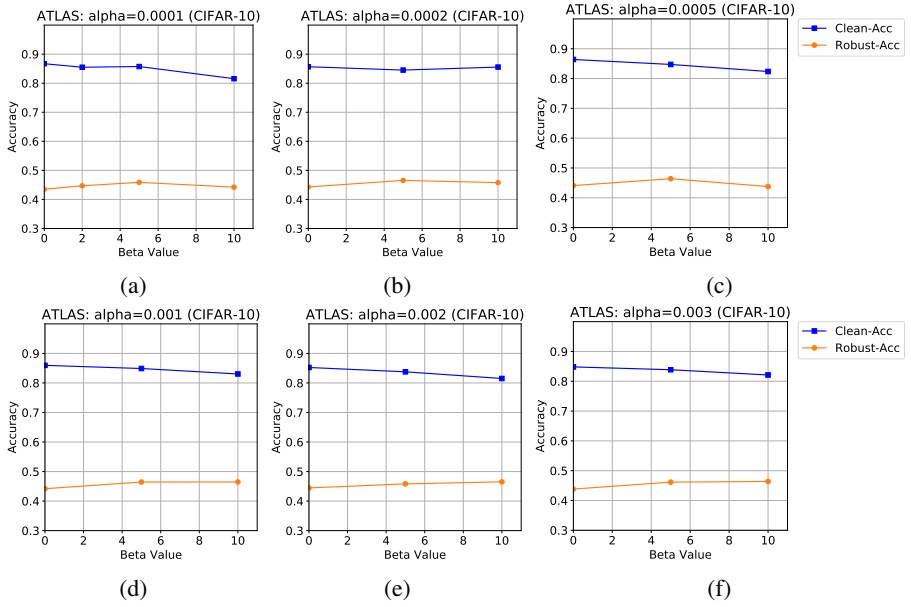

Figure 16: Clean and robust accuracy for one-step ATLAS at different $\alpha, \beta$ on CIFAR-10

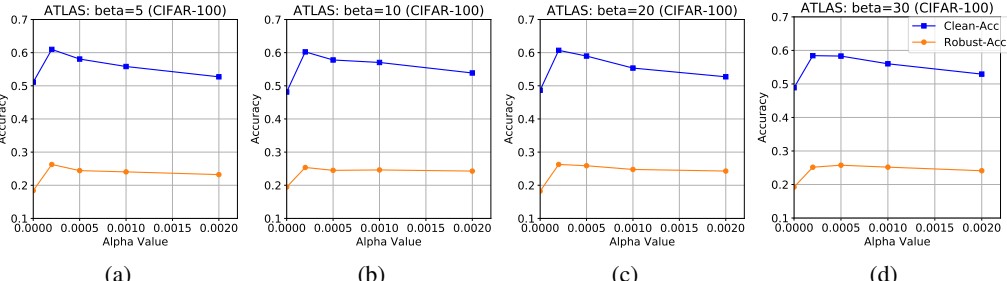

Figure 17: Clean and robust accuracy for one-step ATLAS at different $\alpha, \beta$ on CIFAR-100

