# OpenReview forum: "Improving Local Effectiveness for Global Robustness Training"
_ICLR.cc/2021/Conference — Reject_

### Official Review · AnonReviewer1 · 2020-10-27
**Limited novelty and some concerns regarding the experimental results**

**Rating:** 4
**Confidence:** 5

**Review:**

The paper proposes a new adversarial training scheme, LEAP, to obtain models robust against $\ell_\infty$-bounded adversarial examples. The loss used as the objective minimized during training involves both local and global (wrt the input space) properties of the network. Experiments suggest improved performance compared to single and multi-step standard adversarial training and TRADES.

Pros
1. The proposed method achieves in the reported experiments better results than existing methods, for both single and multi-step adversarial training.
2. The authors provide detailed ablation studies in the appendix.

Cons
1. The final loss used looks like a straightforward combination of TRADES and a regularization term on the norm of the Jacobian matrix at the adversarial points. As mentioned in Sec. C.1, as differences from TRADES, first LEAP uses the adversarial images rather than the clean ones when computing the cross entropy loss, and second it computes the adversarial examples maximizing the cross entropy rather than the KL distance. Both seem minor modifications: the first change should lead in principle to lower clean accuracy (in Table 1 LEAP models have 1-2% lower clean accuracy than TRADES ones on CIFAR-10 and CIFAR-100) and worse trade-off clean vs robust accuracy, while the second one is not discussed in the main paper. Moreover, the regularization of the Jacobian matrix is, as acknowledge in the text, not novel.
2. Although the experimental results seem to favor LEAP, I have some concerns about the hyperparameters used in PGD at training and test time. For the multi-step adversarial training on MNIST, with $\epsilon=0.3$, 20 steps of PGD with step size 0.01 are used (Sec. B), meaning that the $\ell_\infty$-ball cannot be fully explored by the attack. Conversely, on CIFAR-10 and CIFAR-100 a step size of 0.07 > 17/255 is used with and $\ell_\infty$-ball of radius 8/255, which is quite large with a budget of 10 steps. For testing the robustness of the models, the MultiTargeted attack is used: it is a strong attack, but it is given a budget of only 20 iterations, which might be insufficient for the attack to be effective. Increasing the budget for the attacks or using some standard evaluation like that from (Croce & Hein, 2020) would strengthen the results.
3. When evaluating LEAP on cheap adversarial training, considering that the computational cost is roughly 3x that of one-step ADV, also the method from (Andriushchenko & Flammarion, 2020) should be included in the comparison, as it should have a runtime similar to the proposed scheme and it is shown to outperform one-step ADV.

Overall, I think the novelty is very limited, which combined with some concerns about the experiments makes me lean towards giving a negative score.

Croce & Hein, "Reliable evaluation of adversarial robustness with an ensemble of diverse parameter-free attacks"\
Andriushchenko & Flammarion, "Understanding and Improving Fast Adversarial Training"

---
Update after rebuttal

I thank the authors for the response. I think that the revised version improved clarity. The overall impression I have of the paper is still similar to the initial one.

The final proposed loss is reasonable, but just the combination of two existing ones with minor modifications. About this, even in the revised version the authors seem not to discuss and justify (at least in the main part) how the adversarial point $x'$ in computed at training time, which is different from TRADES according to Section C.1.

About the experimental part, I thank the authors for adding the new experiments in Section D. However, the baseline in Table 4 seems a bit weak, at least for 10 steps ADV. For reference, the baseline WRN-28-10 in (Gowal et al., 2020) has robustness under AutoAttack + MultiTargeted close to 51% (I see that here a WRN-28-8 is used, but I wouldn't expect such difference).
Moreover, although a minor concern, the authors didn't address the question about the step size used on MNIST.

Then, I keep my initial score.

Gowal et al., "Uncovering the Limits of Adversarial Training against Norm-Bounded Adversarial Examples"

---

> ### Author Response · Authors · 2020-11-20
> **Thank you for your review, comments and questions.**
>
> 1	Although the final loss may take a similar form of combining TRADES and a Jacobian regularizer, the underlying idea is new and the loss development is completely different from a simple combination. We have made major reorganizations to the paper to better convey the novelty of our framework. Firstly, we introduce a different perspective on robust training loss and propose a new framework for constructing losses. We have included a binary example and figures to better illustrate the intuition and the key idea. Secondly, we develop a novel algorithm based on the framework for fast robust training.
>
> In terms of results of LEAP (We have changed the method name to ATLAS to remove all arguments related to patches) and TRADES in table 1, the difference should not be simply attributed to a trade-off clean vs robust accuracy. For one-step TRADES, we have tested a wide range of beta values. We find that with larger beta values, the robust accuracy is not improved but the clean accuracy suffers. Plots can be found in Appendix F.2. Since a similar level of robust accuracy is obtained for all beta values, we thus picke the model with the highest clean accuracy.
>
> When one-step TRADES is compared against LEAP-g (ATLAS-g in current version), the main difference is that LEAP-g computes cross-entropy loss at weak adversaries in order to maximize the use of adversaries. One-step LEAP-g overall outperforms one-step TRADES in both clean and robust accuracy shows the importance of effective use of weak adversaries in one-step setting.
>
> The key of the local step is to increase the radius of local robust balls. Depending on the problem, various forms of regularizers could be used. An efficient approximation of Jacobian norm is a suitable choice for fast robust training. We believe it is more important to recognize the key underlying idea so different losses can be constructed to best accommodate the problem at hand.
>
> 2	We thank the reviewer for rigorously examining our experimental setup. The step-size for CIFAR-10 and CIFAR-100 should be 0.007. It was a typo.  We make these step size choices to be consistent with the common set-up used in adversarial training studies. There are two main reasons for the budget of 20 steps for Multi-targeted attacks. Firstly, we find that the robust results change negligibly when increasing the number of steps up to 100. Secondly, we have comparatively limited computing resources. Based on the reviewer's suggestion, we have added the AutoAttack evaluation and results are reported in Appendix D.2. We also summarize the results as follows:
>
> Results for AutoAttack Standard Version
>
> Model 			CIFAR-10 	CIFAR-100
>
> Zero-step Jac		30.03%	11.85%
>
> One-step ADV		42.49%	17.17%
>
> One-step TRADES 	37.74%	12.75%
>
> One-step ATLAS 	46.16%	23.42%
>
> One-step TradesJac 	43.16%	17.99%
>
>
>
> Ten-step ADV		47.56%	24.64%
>
> Ten-step TRADES	49.40%	22.78%
>
> Ten-step ATLAS 	49.82%	25.77%
>
>
> 3	We have added the following discussion in the experiment part:
>
> “In addition, we constantly find for one-step TRADES that, after a certain number of epochs, the percentage of the model making incorrect decisions on weak adversaries falls sharply, leading to a sudden drop of more than 10% on validation robust accuracy. This phenomena is referred to as catastrophic overfitting (Wong et al., 2020; Andriushchenko & Flammarion, 2020). We observe that when this behaviour occurs, the approximated value ||J(\boldsymbol{x}')||_{F}^{approx} increases steeply, which is consistent with what has been observed in Andriushchenko et al.(2020). When dealing with catastrophic overfitting is the main concern, Andriushchenko et al.(2020) argue the key is to increase local linearity and they introduce a regularizer to maximize gradient alignment for points within the perturbation ball. In our case, the issue could be similarly resolved by increasing the coefficient $\alpha$ to encourage local linearity. However, since the goal of this work is to achieve high robust accuracy fast, we do not restrict ourselves to models without catastrophic overfitting only, which are likely to compensate robust accuracy for stability. Instead, we use early termination and consider a wider range of models. We mention that the same phenomenon is observed on one-step ADV and on one-step ATLAS when $\alpha$ is small but less frequently. More discussions on catastrophic overfitting can be found in Appendix E. ”
>
> We point out that Andriushchenko et al.(2020) is concurrent work. Nonetheless, we have included a discussion to help contextualise our approach.

---

### Official Review · AnonReviewer3 · 2020-10-29
**Official Blind Review #3**

**Rating:** 5
**Confidence:** 5

**Review:**

Summary:
The authors developed a novel robust training algorithm LEAP to focus on the effective use of adversaries. The proposed method improves the model robustness at each local patch and combines these patches through a global term, achieves overall robustness. The authors showed by maximizing the use of adversaries, they achieved high robust accuracy with weak adversaries. Furthermore, when trained with strong adversaries, the proposed method matches with the current state of the art on MNIST and outperforms them on CIFAR-10 and CIFAR-100.


Comments:

1 . The authors’ main idea is to promote local patch robustness combined with global robustness. While the local patch robustness is further related to the Jacobian norm, the global robustness is exactly the same as TRADES’ regularization term. My major concern is that the proposed method still does not have a very convincing intuition that why the local patch robustness term or combining the two terms helps. In particular, the LEAP_g algorithm is actually very close to TRADES, why it could achieve better robustness. The authors might want to add more explanations/demonstrative experiments to show that.


2 . In eq (5), what does it mean by decision boundary = 0? Also in eq (6) what is standard computation? I assume the authors refer to first-order Tylor expansion? The authors need to reorganize the presentation of this part to make everything clear. And equation (5)/(6) is actually useless since finally, the authors only rely on Proposition 1 to promote the local patch robustness term. For the approximation of the Jacobian norm, the authors might want to briefly introduce some details on how the approximation is done.

3 . The following work also performs robust training directly on x’ instead of x,

"Improving adversarial robustness requires revisiting misclassified examples." ICLR (2019).

The authors might also want to comment on it.

4 . Notice that even adversarial training based algorithms could cause obfuscated gradient problem, therefore, it might be a good idea to further evaluate model robustness via totally gradient-free methods, such as hard-label attacks

“RayS: A Ray Searching Method for Hard-label Adversarial Attack” KDD (2020)

In order to make the experimental results more convincing.


5 . In section 4.1, “Since l∞ norm is equivalent to l2 norm”. It is fine to only present L2 norm analysis but this statement is not appropriate and may cause confusion. In cases where the data dimension N is large, the difference between the two settings could also be drastically different.

---

> ### Author Response · Authors · 2020-11-20
> **Thank you for your review, comments and questions.**
>
> 1	To address the reviewer’s concern, we have reorganized the paper. We first introduce our different perspective and our novel framework for constructing losses. We have added a binary example and illustrative figures. Hopefully, with these modifications and additions, the intuition and key underlying ideas can be better conveyed. The proposed algorithm is based on the framework and is specifically designed for fast robust training. ATLAS-g (We have renamed LEAP as ATLAS to remove confusing patch arguments) makes a more effective use of weak adversaries by computing cross-entropy loss at them instead of natural images. This modification is important in the one-step robust training setting. More details are provided in Appendix C.1.
>
> 2	 We have rephrased the part introducing the Jacobian regularizer. We have replaced decision boundary with
> “The boundary hyper-surface separating classes c and c’ consists of points x^b satisfying f_c(x^b; \theta) - f_c’(x^b; \theta).”
> The standard computation is referring to the formula for computing the distance between a point and a hyper-plane. We have added all these details. Equations (5)/(6) (Equation (9)/(10) in the current version) are important, because we need them to define the distance d. The goal is to maximize d for larger robust ball radius. We do so by maximizing the lower bound of d, provided in proposition 1 (proposition 2 in the current version).  A direct application of random projection is used to approximate the Jacobian norm. We have added this detail.
>
> 3	We have added the work in the related work section.
>
> 4	We thank the reviewer for the suggestion. We have included the results of RayS in Appendix D.2.  We also provide the result in the following:
>
> Results for RayS with 40000 queries on 1000 randomly selected images
>
> Model 			CIFAR-10 	CIFAR-100
>
> Zero-step Jac		33.50%	16.70%
>
> One-step ADV		50.20%	19.90%
>
> One-step TRADES 	45.10%	19.50%
>
> One-step ATLAS 	52.70%	28.60%
>
> One-step TradesJac 	50.20%	22.60%
>
>
> Ten-step ADV		54.40%	29.80%
>
> Ten-step TRADES	54.00%	28.20%
>
> Ten-step ATLAS 	53.20%	30.60%
>
> 5	We have removed the sentence. We present all our analyses in L2 norm with a mention of the generalizability of our analyses to other norms.

---

### Official Review · AnonReviewer4 · 2020-10-29
**Initial review**

**Rating:** 5
**Confidence:** 5

**Review:**

In this paper, the authors introduce a novel technique (called LEAP) that improves model robustness at local patches (around an adversarial example) and combines "local patches" to get global robustness. Their approach works better than other techniques when using a "weak" adversary.

Overall, the paper is well structured. However, it is a bit confusing at points, and sometimes not clearly motivated. The experiments seem fair and demonstrate that the proposed approach is better than other state-of-the-art techniques.

1) I find the notion of local patches difficult to understand, and the paper seems to go through many hoops to justify a rather simple idea (simple ideas are good). This paper reminds me a lot of [1] and the loss presented could be justified through other means.
2) Talking of [1] (although this is not necessary due to short amount of time between NeurIPS acceptance and ICLR submission), I'd appreciate a discussion on the differences.
3) When mentioning PGD, cite [2] (for BIM). When mentioning FGSM, cite [3].
4) When W^x and b^x are initially introduced, it is unclear what they are representing. Is that a first order Taylor expansion around x?
5) A few sentences seem useless and sometimes the language is hand-wavy: e.g., "it is highly likely the volume of patch P_x is non-zero".
6) The whole paragraph before section 4.1 does not seem to be useful for the rest of the paper.
7) It is unclear in Sec 4.1 whether proposition 1 is part of Hoffman et al. (2019).
8) It took me a while to understand the A(J(x)) notation. A() seems to be a function that takes J(x) as input. Consider directly using the Frobenius norm |J(x')|_F here and mention that it can be approximated.
9) Re-ordering the final loss a bit, we have L_leap = l(x') + \beta * KL(...) + \alpha * |J(x')|_F. The first two terms are reminiscent of TRADES (except that they are applied to x') and the last term acts as a regularization term (similar to one used in [1] or [4]). Could the experiments also include an alternative loss:  l(x) + \beta * KL(...) + \alpha * |J(x)|_F (combining TRADES and zero-step JAC).
10) When mentioning Untargeted attack with margin loss, cite [5]. When mentioning Multi-targeted, cite [6].
11) The authors seem to use beta = 4 for TRADES. However, TRADES seem to work better with beta = 6. In particular, training a WRN-28-10 using TRADES with 10 steps of PGD on CIFAR-10 should reach 51-52% robust accuracy, the authors get 49.66% for a WRN-28-8. Can the authors explain the gap?
13) The authors should also compare with "Adversarial training for free!" [7]

Other comments:

A) Qin et al. [4] is lumped with techniques that requires a strong adversary. However, Qin et al. show that only 2 PGD steps are necessary.
B) Table 1 could include the number of steps rather than the vague "multi-step" wording.
C) The Discussion seems more like a Conclusion.

[1] https://arxiv.org/pdf/2007.02617: Understanding and Improving Fast Adversarial Training
[2] https://arxiv.org/pdf/1607.02533: Adversarial examples in the physical world
[3] https://arxiv.org/pdf/1412.6572: Explaining and Harnessing Adversarial Examples
[4] https://arxiv.org/pdf/1907.02610: Adversarial Robustness through Local Linearization
[5] https://arxiv.org/pdf/1705.07263: Adversarial Examples Are Not Easily Detected: Bypassing Ten Detection Methods
[6] https://arxiv.org/pdf/1910.09338: An Alternative Surrogate Loss for PGD-based Adversarial Testing
[7] https://arxiv.org/pdf/1904.12843: Adversarial Training for Free!

---

> ### Author Response · Authors · 2020-11-20
> **Thank you for your review, comments and questions.**
>
> 1	We have removed all arguments related to patched and made some major re-organizations to the paper. In the current version, we first introduce our novel perspective on robust training and propose a new framework for constructing losses. To better convey our key ideas, we have added a binary example and illustrative figures. Then, we go into the details of how the novel algorithm is developed based on the framework. Hopefully, with these modifications and additions, we can better convey the intuition of our original ideas.
>
> 2	We have added the following discussion in the experiment part:
>
> “In addition, we constantly find for one-step TRADES that, after a certain number of epochs, the percentage of the model making incorrect decisions on weak adversaries falls sharply, leading to a sudden drop of more than 10% on validation robust accuracy. This phenomena is referred to as catastrophic overfitting (Wong et al., 2020; Andriushchenko & Flammarion, 2020). We observe that when this behaviour occurs, the approximated value ||J(\boldsymbol{x}')||_{F}^{approx} increases steeply, which is consistent with what has been observed in Andriushchenko et al.(2020). When dealing with catastrophic overfitting is the main concern, Andriushchenko et al.(2020) argue the key is to increase local linearity and they introduce a regularizer to maximize gradient alignment for points within the perturbation ball. In our case, the issue could be similarly resolved by increasing the coefficient $\alpha$ to encourage local linearity. However, since the goal of this work is to achieve high robust accuracy fast, we do not restrict ourselves to models without catastrophic overfitting only, which are likely to compensate robust accuracy for stability. Instead, we use early termination and consider a wider range of models. We mention that the same phenomenon is observed on one-step ADV and on one-step ATLAS when $\alpha$ is small but less frequently. More discussions on catastrophic overfitting can be found in Appendix E. ”
>
> 3	The citation for PGD has been added and the citation for FGSM was already included.
>
> 4	Here, we are using the fact that the neural network is piecewise linear when ReLU activation is used. We have rephrased sentences to “We first assume that the neural network f contains ReLU activation for the sake of clarity. This implies that the neural network is piecewise linear” for better clarification.
> The possibility of writing f(x;\theta) = W^x x + b^x is a consequence of the network being piecewise linear. In other words, W^x and b^x represent the parameters of the linear part of the function at x.
>
> 5	These sentences have been removed
>
> 6	The purpose is to introduce the overall underlying idea that motivated the development of the loss. In the current version, we illustrate our key original ideas through the introduction of the new framework. Then we focus on the development of the loss, which is based on the framework.
>
> 7	As indicated in the paper, proposition 1 (proposition 2 in the current version) is from Jakubovitz & Giryes (2018). We use the proposition to explain why we want to use Jacobian norm. Hoffman et al. (2019) focus on estimating Jacobian norm efficiently.
>
> 8	We have replaced A() with ||J(x)||^{approx}_F to be clearer.
>
> 9	We have added the loss and named it as TradesJac. Results and analyses are in Appendix D.1.
>
> 10	Citations for Untargeted attack and Multitargeted attack are added.
>
> 11	We believe it is mainly due to the fact that different architectures are used. For fair comparison, we have run Trades with beta ranging from 0 to 10. We used beta=4, because it gives the best validation accuracy.
>
> 12	It has been demonstrated in [1] that one-step ADV gives better performance in fast robust training setting. Therefore, we have chosen to compare against the stronger baseline, namely one-step ADV.
>
> For other comments:
>
> 1	We thank the reviewer for pointing this out. We have moved Qin et al. to the paragraph on weak adversaries.
>
> 2	We have changed “multi” to the exact number.
>
> 3	We have modified the discussion. The current version is
>
> “We have adopted a different perspective and proposed a new framework for constructing losses that allow more effective use of adversaries. Specifically, based on the framework, we introduce a novel algorithm ATLAS for fast robust training. ATLAS can be used as an initialisation technique for other complicated tasks. For instance, model trained with ATLAS can be employed as the starting point for layer-wise adversarial training in improving certified robustness. Apart from fast robust training, we believe other losses could be constructed from the framework to accommodate different problems. We leave the explorations of potential applications to various robust training setting to future research.”
>
> [1] Eric Wong, Leslie Rice, and J. Zico Kolter. Fast is better than free: Revisiting adversarial training. The International Conference on Learning Representations, 2020.

---

### Official Review · AnonReviewer2 · 2020-10-30
**Good empirical results but there are concerns about the novelty and clarity of the provided justifications**

**Rating:** 5
**Confidence:** 4

**Review:**

**Summary:**
The paper proposes a new way to combine existing techniques for improving adversarial robustness: adversarial training, Jacobian regularization and TRADES consistency loss. The obtained results on three datasets are better than the baselines which rely on adversarial training, Jacobian regularization or TRADES separately.

**Pros:**
- Good empirical results.
- Extensive empirical evaluation on MNIST, CIFAR-10, CIFAR-100.
- Ablation study for the components of the method.

**Cons:**
- My main concern is the novelty of the proposed approach. The paper proposes to use adversarial training together with the TRADES-loss (with a reversed KL-divergence) and a variant of approximate gradient penalization at an adversarial point. All these approaches existed before but now just all combined together.
- Additional concern is that the method requires 2 additional hyperparameters alpha and beta compared to usual adversarial training that doesn’t lead to any additional hyperparameters. I think this concern is especially relevant since the method is proposed to be useful for fast adversarial training (i.e. with one-step adversarial examples).
- Another concern is the clarity of the presentation. It was really hard to grasp what are the sets $P_x$, $P_x’$, $S_\theta(x)$, $S_{\theta^1}(x)$, $S_{\theta^2}(x)$ and relations between them. Maybe it would be better to clarify it with a picture / diagram. Currently, the theoretical part doesn’t seem to be clear or convincing enough to me.
- It would be more insightful if one can provide a clear discussion on how and why the proposed method mitigates the catastrophic overfitting problem (similarly to these recent papers [Li et al. (2020)](https://arxiv.org/pdf/2006.03089.pdf) and [Andriushchenko et al. (2020)](https://arxiv.org/pdf/2007.02617.pdf) which focus on overcoming this problem). There are some mentions

**Minor suggestions**
- For me, it seems to be a bit misleading to call a subset of the input space a patch given that in the literature, patches are mostly referred to perturbations on the image plane (e.g. as in [Adversarial Patch paper](https://arxiv.org/abs/1712.09665)).
- Contributions 2 and 3 at the top of page 2 are nearly duplicates.
- “we **take care** of local balls at various points” -- imprecise (what it means to take care in this context?) and a bit informal language.
- Equation (6): in the denominator it should be f_c’ instead of f_c in the second term.
- Page 5: “as we evaluate it at an adversary x’ instead of the image x” -- it’s not yet clear what “an adversary x’” means, in particular with which method this point is obtained, is it epsilon-bounded (I suppose yes but this becomes clear only much later in the text) or not. Would be good to clarify this.
- Page 7: “validation set robust accuracy guided learning rate scheduler” -- consider splitting this phrase which is too complicated.
- Page 7: “natural images will **prevail** when weak adversaries are used” -- meaning is unclear.

**Score:**
5/10 because of the concerns about the novelty and clarity of the provided justifications of the method.

---

> ### Author Response · Authors · 2020-11-20
> **Thank you for your review, comments and questions.**
>
> 1	We have taken a different perspective of robust training and proposed a new framework for constructing losses. While some of the losses are similar to those proposed in the literature, the differences play a key role in obtaining the state of the art results as shown by our thorough ablation study.
>
> 2	Although two parameters are required, as shown in Appendix F, similar robust accuracy performances can be achieved by a wide range of parameters values. This makes parameter tuning computationally efficient.
>
> 3	We appreciate the reviewer’s suggestion and have made some major changes to our paper in order to better convey our ideas. We have now first introduced the new loss framework to illustrate our different perspective on robust loss and novel ways for constructing losses. To facilitate the discussion, we have added a binary example, accompanied by illustrative figures. Then, we go through the development of our loss algorithm based on the framework.
>
> 4	We have added the following discussion in the experiment part:
> “In addition, we constantly find for one-step TRADES that, after a certain number of epochs, the percentage of the model making incorrect decisions on weak adversaries falls sharply, leading to a sudden drop of more than 10% on validation robust accuracy. This phenomena is referred to as catastrophic overfitting (Wong et al., 2020; Andriushchenko & Flammarion, 2020). We observe that when this behaviour occurs, the approximated value ||J(\boldsymbol{x}')||_{F}^{approx} increases steeply, which is consistent with what has been observed in Andriushchenko et al.(2020). When dealing with catastrophic overfitting is the main concern, Andriushchenko et al.(2020) argue the key is to increase local linearity and they introduce a regularizer to maximize gradient alignment for points within the perturbation ball. In our case, the issue could be similarly resolved by increasing the coefficient $\alpha$ to encourage local linearity. However, since the goal of this work is to achieve high robust accuracy fast, we do not restrict ourselves to models without catastrophic overfitting only, which are likely to compensate robust accuracy for stability. Instead, we use early termination and consider a wider range of models. We mention that the same phenomenon is observed on one-step ADV and on one-step ATLAS when $\alpha$ is small but less frequently. More discussions on catastrophic overfitting can be found in Appendix E. ”
>
>
> For the minor suggestions:
>
> 1	We have followed the reviewer’s suggestion and removed all arguments involving patches.
>
> 2	We have modified our contributions to make them more clearer. They should be contributions 3 and 4 now. Contribution 3 is focusing on the effective use of weak adversaries of our algorithm. Contribution 4 mentions that although our algorithm is slightly more expensive, it still allows efficient robust training.
>
> 3,4	Imprecise language is removed. The typo is fixed.
>
> 5	We have added the following sentence for a clarification:
> “In the fast robust training setting, x’ is a weak adversary obtained through one-step FGSM.”
>
> 6	We have splitted the phrase to “a learning rate scheduler, which is guided by the robust accuracy on the validation set”.
>
> 7	We have removed the sentence and changed the explanation to:
> “Since it uses the natural image x to compute cross-entropy loss and as the base distribution in the KL regularizer, TRADES cannot use adversaries effectively when they are weak”.

---

> > ### Comment · AnonReviewer2 · 2020-11-24
> > **The main concerns remain, keeping the score**
> >
> > Thank you for the answer.
> >
> > I appreciate providing a clearer picture to explain the motivation behind the proposed method and a discussion of the recent paper "Understanding and Improving Fast Adversarial Training". However:
> > * I'm not fully convinced by the insensitivity to the hyperparameters -- in my opinion, Fig. 16 shows that the optimal beta is different for different values of alpha, and this difference can be crucial, especially since the paper argues about the improvements on the order of 0.5%-3% compared to the existing baselines.
> > * The motivation behind local vs global robustness (albeit now with a clearer presentation) sounds handwavy and heuristical to me. I would suggest to clearly formalize it and give some quantitative evaluation of both types of robustness.
> >
> > To summarize, I would like to keep my original score, especially after reading the concerns raised by other reviewers. In particular, I agree with Reviewer 1 that overall the method is just a combination of existing training objectives (TRADES + Jacobian regularization) which was also my impression as well.

---

### Author Response · Authors · 2020-11-20
**To All Reviewers**

We thank all reviewers for your insightful comments and constructive suggestions.

1.   Several reviewers mention that the underlying idea illustrated through patches is difficult to understand. We truly appreciate the feedback. We thus have made major organization changes to the paper and removed all arguments related to patches. Hopefully, with these modifications, we can effectively communicate our new ideas for constructing robust losses and the development of our novel algorithm. As a result, we have renamed our method: Adversarial Training via LocAl Stability (ATLAS).

2.    In terms of reorganization, we start by introducing the novel perspective we take on robust training and proposing a new framework for constructing robust losses. We have used a binary example and added figures for better illustration of the underlying idea. We then introduce our novel algorithm for fast robust training based on the framework.

3.    Several reviewers mention the Neurips 2020 paper by Andriushchenko et al.. We point out that the paper is concurrent work (as acknowledged by AnonReviewer4). Nonetheless, we have added a discussion that highlights the difference between the two frameworks.

---

### Decision · Program_Chairs · 2021-01-07
**Final Decision**

**Decision:**

Reject

**Comment:**

This paper presents a new method of employing some existing techniques to improve robustness, which was verified through experiments. According to the reviewers’ comments and the authors’ responses to these comments, the reviewers generally appreciate the authors’ effort in properly improving and clarifying the proposed method. However, their major concerns still rely on the novelty of this paper, which is identified as a combination of some existing techniques. In addition, the proposed method at its current status still contains some un-convincing points. Hence, the paper is recommended rejected.